# Genomic and immune landscape Of metastatic pheochromocytoma and paraganglioma

The mechanisms triggering metastasis in pheochromocytoma/paraganglioma are unknown, hindering therapeutic options for patients with metastatic tumors (mPPGL). Herein we show by genomic profiling of a large cohort of mPPGLs that high mutational load, microsatellite instability and somatic copy-number alteration burden are associated with *ATRX/TERT* alterations and are suitable prognostic markers. Transcriptomic analysis defines the signaling networks involved in the acquisition of metastatic competence and establishes a gene signature related to mPPGLs, highlighting CDK1 as an additional mPPGL marker. Immunogenomics accompanied by immunohistochemistry identifies a heterogeneous ecosystem at the tumor microenvironment level, linked to the genomic subtype and tumor behavior. Specifically, we define a general immunosuppressive microenvironment in mPPGLs, the exception being PD-L1 expressing *MAML3*-related tumors. Our study reveals canonical markers for risk of metastasis, and suggests the usefulness of including immune parameters in clinical management for PPGL prognostication and identification of patients who might benefit from immunotherapy.

Pheochromocytomas and paragangliomas (PPGLs) are rare and highly heterogeneous neuroendocrine tumors, associated with mutations in one of the at least 20 driver genes related to the disease[1]. Up to 20% of PPGLs are metastatic (mPPGLs), with a heterogeneous 5-year overall survival rate that ranges from 40 to 77%[2]. Although some genotype-phenotype correlations have been defined for patients with PPGLs, disease management remains challenging due to a variable clinical behavior, lack of accurate markers of metastasis risk and inefficient therapeutic standards for metastatic stage[3,4]. Moreover, a poor understanding of the underlying mechanisms driving metastasis hinders the development of an efficient therapeutic strategy for mPPGLs.

The genomic landscape of PPGLs has been poorly described and little is known about the role that secondary events play in the progression of the disease. So far, only two large-scale multi-omic studies reporting the genomic characteristics of PPGLs have been published[5,6]. These studies established the molecular classification of PPGLs in three genomic subtypes: pseudohypoxic, kinase signaling, and Wnt-altered. However, these previous studies included few metastatic cases.

Moreover, the tumor microenvironment (TME) has been shown to play a major role in the prognosis and therapeutic stratification of other tumor types[7]. However, the TME remains largely unexplored in PPGLs and may be pivotal for therapy success.

In this large-scale study, we performed a comprehensive characterization of the genomic landscape of mPPGLs and interrogated the immune compartment of these tumors. We compared the results from our series, enriched in metastatic cases, with the TCGA PPGL cohort[5], which is mainly composed of non-metastatic tumors. Using this strategy, we defined the mutational architecture of mPPGL, identified transcriptional alterations linked to metastasis-related processes, proposed prognostic markers, and outlined the immune infiltration profile in PPGL.

## Results

We performed genomic profiling of 156 PPGLs from 128 unrelated patients: whole-exome sequencing (WES) in 87 paired germline–tumor samples and RNA sequencing (RNA-Seq) in 114 tumor samples

✉e-mail: bcalsina@cnio.es; mrobledo@cnio.es

(hereinafter, CNIO cohort; Fig. 1a; Supplementary Table 1). Seventy-five of these patients had metastatic disease, detected at initial diagnosis or during follow-up, and accounted for 99 metastatic samples included in the study (either primary tumors, relapses or metastases). We also included in the study data of 175 PPGLs from 174 unrelated patients from the PPGL TCGA project: WES data from 174 paired germline–tumor samples and RNA-Seq data from 150 tumors (from now on, TCGA cohort). By a comprehensive analysis of these data, we deciphered the mutational, copy number alteration (CNA),

transcriptomic and immune landscapes of mPPGLs (Fig. 1b). Tumors were classified according to their genotype/ transcriptional profile into the previously defined PPGL genomic subtypes[5].

## Mutational landscape of metastatic pheochromocytoma and paraganglioma

We identified somatic single-nucleotide variations (SNVs) and small insertions/deletions (INDELs) in 87 paired germline-tumor samples with available WES data (mean depth of 105×; 12 tumors from 11

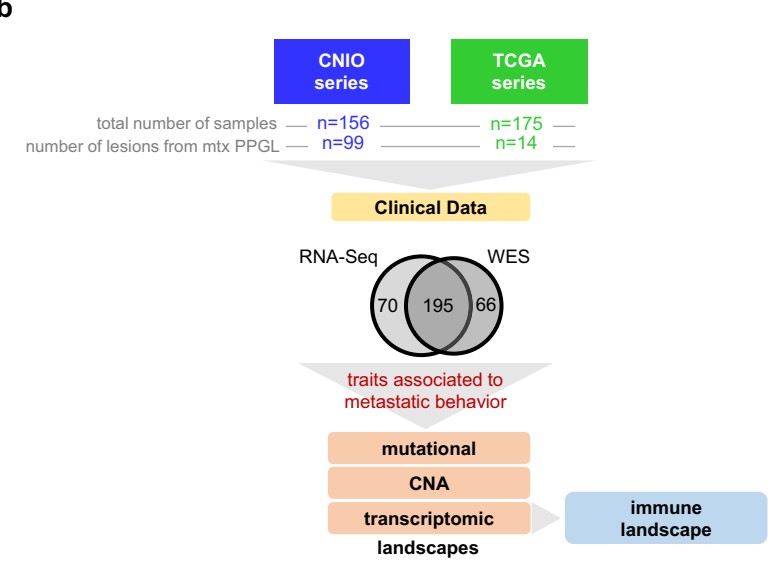

**Fig. 1 | Series description and study workflow. a** Characteristics of the CNIO cohort used for WES and RNA-Seq analysis. Characteristics shown are: platform available (WES and RNA-Seq), preservation of material (FFPE or frozen), tumor type (non-metastatic, aggressive, metastatic, relapse or metastases), primary tumor and metastasis location, sex, age at diagnosis, genomic subtype and genotype. NA: information not available. **b** General overview of the study. See also Supplementary Table 1 and Supplementary Fig. 1.

patients were discarded due to poor quality; see methods section). Data from 169 TCGA patients (174 samples) were added for the analyses (Supplementary Fig. 1). To note, for 20 patients (16 from CNIO cohort and 4 from TCGA cohort), we had available more than one tumor sequenced by WES (range, 2–7).

WES data analysis revealed 5618 somatic variants across 261 samples. From those, 3641 had a variant allele frequency (VAF) > 10% and 2227 were missense, 140 truncating, 127 affected start/stop codons, 52 in-frame deletions or insertions, 66 altered canonical splice sites, and 1029 were classified as 'other' (including synonymous variants); 3.2% of the variants were classified as high impact consequence variants and affected cancer genes (from COSMIC Cancer Gene Census or OncoKB).

WES data was used to calculate the tumor mutational burden (TMB) and microsatellite instability (MSI) scores of each tumor (Fig. 2a). The TMB (based on true-somatic coding events with VAF > 0.1; category 5 in Supplementary Fig. 2) varied between cases, with a median of 11 (range, 0–142), in agreement with previous PPGL studies[5,6,8,9], and much lower than in other cancer types[10]. Two of the PPGL studies suggested an association between higher TMB and metastatic behavior[5,9]. In this study, we confirmed this observation and also detected that TMB increased from non-metastatic primary tumors (median number of variants: 8; range 2–25) to primaries of mPPGLs (17.5; 1–54) to metastases (24; 0–142) (Fig. 2b). We also observed a significantly different TMB among genomic subtypes, with the Wnt-altered subtype having the highest values, regardless of tumor behavior (Supplementary Fig. 3a, b). Although there is scarce information about MSI in PPGLs[11–13], it is known that MSI contributes to the tumorigenic hypermutated phenotype in other cancer types[14]. In our study, we observed differences in the MSI score according to the clinical behavior (non-metastatic tumors median: 0.13, range 0.11–0.18; primary metastatic tumors: 0.16, range 0.10–0.22; and metastases median: 0.18, range 0.11–0.25) (Fig. 2c). Moreover, the MSI score and the TMB exhibited a moderately significant correlation (Fig. 2d), indicating that the higher TMB observed in metastatic tumors could be linked to a higher MSI status. Both features were associated with risk of metastasis development in univariate and multivariate analyses (including as covariates sex, mutations in Krebs cycle susceptibility genes[15] and ATRX/TERT alterations), suggesting they are independent predictors of metastatic risk (Table 1).

To note, both higher TMB and MSI score were also associated with a shorter time to progression (TTP) (Table 1; Fig. 2e, f; Supplementary Fig. 3c). In agreement, we observed a significantly higher TMB in larger tumors, and in those with >5% Ki67 positive cells and high MKI67 expression levels (Fig. 2g–i); no association was detected between the MSI score and the tumor volume or the % of Ki67 positive cells (Supplementary Fig. 3d, e), probably due to the low number of tumors included in this analysis, but a significant association between the MSI score and the MKI67 expression levels was identified (Supplementary Fig. 3f).

TMB in PPGLs also correlated with the age of the patients at surgery (Fig. 2j), feature that has already been shown for many cancers[16]. In fact, it has been described that the age-dependent increase in somatic variants that occurs naturally in tissues, increases the probability of mutating driver genes, making cancer a disease of ageing[17].

Without considering the 10 main somatic PPGL gene drivers represented in our cohort (HRAS, FGFR1, CSDE1, MAML3, IDH1, RET, VHL, NF1, MAX, and EPAS1), ATRX was found mutated in the tumors of 3.4% patients (Supplementary Table 2) and was established as the only gene significantly mutated in metastatic tumors (Fisher's test, $P = 2.9 \times 10^{-4}$), as previously reported[8]. Since TERT has been linked to mPPGL[18], we also explored TERT alterations (overexpression, amplification, promoter mutations and hypermethylation; see methods section) in a subset of cases, and confirmed that TERT alterations are events associated with mPPGLs (Supplementary Fig. 3g; Fisher's test, $P = 8.4 \times 10^{-7}$). We validated that ATRX and TERT alterations accumulate

mainly in SDHB/FH pseudohypoxic tumors[19], but also found a high frequency of these events in Wnt-altered tumors (Fig. 2k). Moreover, ATRX/TERT-altered samples presented higher TMB and MSI score than non-altered ones (Fig. 2l, m), a significantly higher MIK67 expression (Supplementary Fig. 3h) and a tendency to higher Ki67 positive cells and bigger tumors (Supplementary Fig. 3i, j).

## Somatic copy-number alteration profile in metastatic pheochromocytoma and paraganglioma

We analyzed the somatic CNA (SCNA) profiles of the CNIO and TCGA cohorts applying FACETS to the germline-tumor matched WES data (Fig. 3a). Higher SCNA burden, measured by the number of SCNA events, was observed in more aggressive tumors (Fig. 3b), with metastatic samples (primaries and metastases from metastatic patients) exhibiting the highest number of SCNA events (median: 15, range: 1–48 and median: 29, range: 5–65, respectively) in comparison to non-metastatic tumors (median: 8, range: 0–41). In addition, a tendency towards shorter time to progression was observed in patients with tumors with high SCNA burden (Supplementary Fig. 4a). Similarly to the TMB, SCNA burden was also higher in ATRX/TERT-altered tumors (Fig. 3c). However, in the case of SCNA burden, no major differences were observed between tumors of different genomic subtypes (Supplementary Fig. 4b).

Genome doubling was observed in 14.5% of primary metastatic tumors whereas only 5% of the non-metastatic tumors duplicated their whole genome ($P = 0.023$, two-tailed Fisher test). Contrary to that reported by Fishbein et al.[5], this tendency was not linked to its genomic subtype ($P = 0.38$, two-tailed Freeman–Halton test).

The most frequent (>25% of all tumors) arm-level deletions affected 1p, 3p, 3q, 11p, 11q, 17p, 22p, and 22q chromosome arms. Arm-level chromosome gains were less common, mostly involving 1q and 7p, as previously reported for PPGLs[6]. Specifically, we observed 1p loss and 1q gain in SDHB tumors; 3p, 4q and the whole chromosome 11 loss in VHL-; 6q loss in HRAS-; chromosome 17 loss in NF1-related tumors. Whole chromosome gains were also detected, chromosome 4 in MAML3- and chromosome 10 in RET-cases. Although 11p deletion is a common event in PPGLs, this SCNA was not observed in tumors harboring oncogenic HRAS mutations (FDR < 0.05). As already shown[5,6], SDHB-, VHL- and NF1-related tumors deleted genomic regions where their specific locus is placed (1p, 3p, and 17q, respectively). MAML3- and RET-related tumors also exhibited gains in chromosomes 4 and 10, comprising MAML3 and RET, respectively.

We identified four altered arm-level SCNA differing between metastatic and non-metastatic primary tumors (Fig. 3a). These included gains in 1q, 5p, and 5q, and deletion of 3q. Whole chromosome 5 gain was also associated with shorter TTP (Fig. 3d), finding which is in line with the fact that TERT is located at 5p15.33. This association remained statistically significant after multivariate Cox hazard regression analysis using sex, mutations in Krebs cycle susceptibility genes, TMB and MSI score as covariates (HR = 2.56, 95% CI = 1.15–5.70, $P = 0.021$), indicating its independent prognostic value.

## Transcriptomic landscape of metastatic pheochromocytoma and paraganglioma

We selected a signature of 26 genes differentially expressed between primary tumors of patients with and without metastatic disease ($n = 54$ and $n = 176$, respectively) and independent of genomic subtypes (Fig. 4a). After expression dichotomization and multivariate analysis using genomic subtype as a covariate, 25 out of the 26 genes were also associated with risk of metastasis ($P < 0.005$; Fig. 4b) and time to progression ($P < 0.05$; Fig. 4c). To validate these findings, we re-analyzed an independent cohort of PPGLs[20] and confirmed a significant differential expression between the two groups for 17 of the 25 selected genes (Supplementary Fig. 5a), including 14 that demonstrated a value for stratifying patients according to metastatic risk

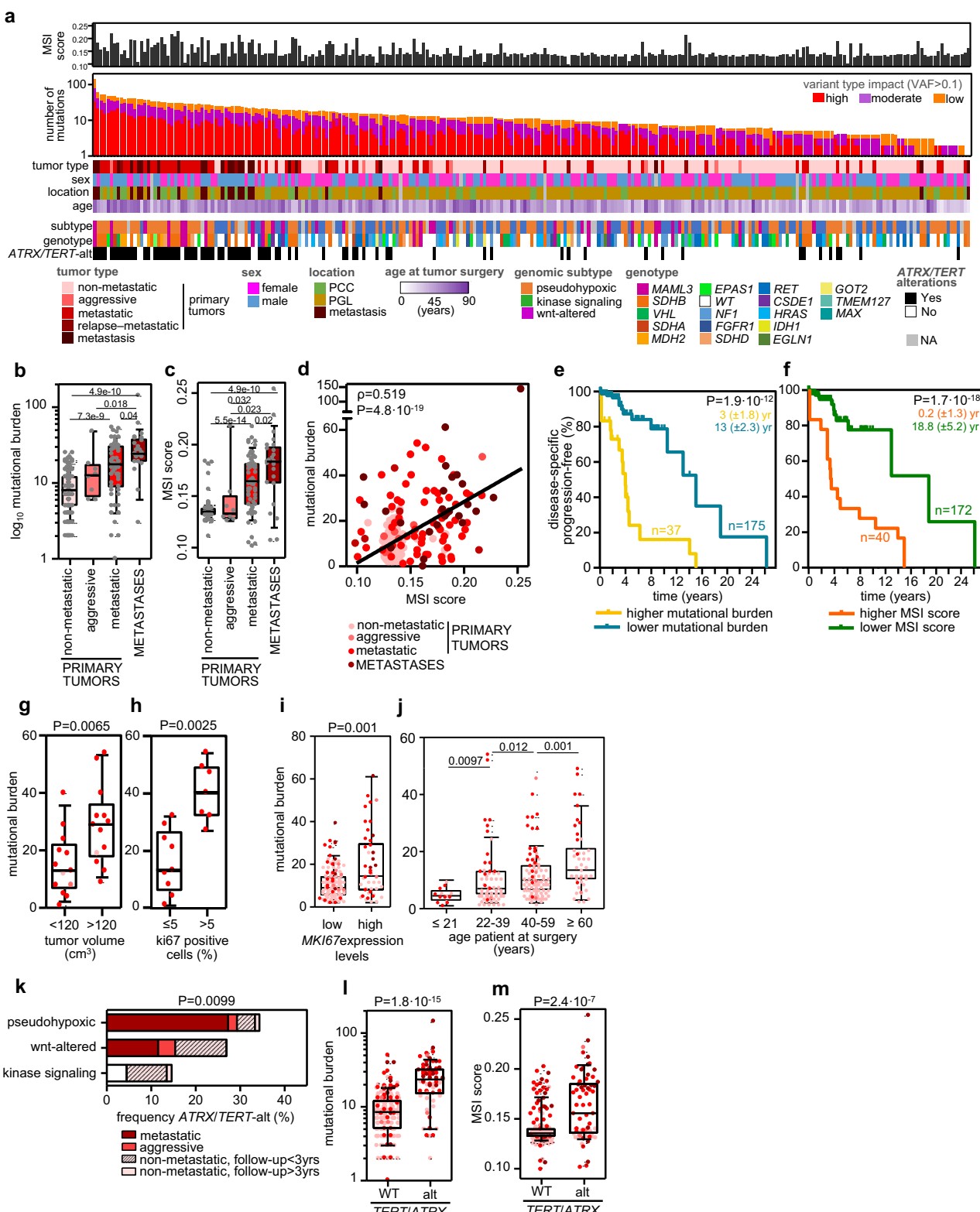

(*C5orf49*, *DNASE1L3*, *HMGB3*, *RRM2*, *CDT1*, *TTC9*, *CCNB2*, *IQGAP3*, *FAM83H*, *CDK1*, *PBK*, *NETO2*, *CSMD2* and *PMAIP1*) (Supplementary Fig. 5b). Functional enrichment analysis of this gene signature using STRING v11[21] indicated a significant overrepresentation (PPI enrichment $P = 3.08 \times 10^{-7}$) of genes related to mitotic cell cycle, G1/S-specific transcription, p53 signaling and DNA damage response (Supplementary Fig. 5c). *CDK1*, the only gene included in the 4 enriched gene sets, displayed the most significant association with metastatic risk in a

combined analysis that included both cohorts (OR = 9.01, 95% CI = 5.17–15.71, $P = 8.5 \times 10^{-15}$; Supplementary Fig. 5d; Fig. 4d); furthermore, high expression of *CDK1* showed a strong association with shorter TTP, regardless of the tumor genomic subtype (HR = 4.15, 95% CI = 2.32–7.24, $P = 1.1 \times 10^{-6}$; Fig. 4e). A multivariate Cox hazard regression analysis including the proposed prognostic molecular markers (chromosome 5 gain, higher TMB and MSI score) as covariates, pinpointed *CDK1* high expression as an additional and

**Fig. 2 | Tumor mutational burden (TMB), microsatellite instability (MSI) and *ATRX/TERT* alterations in mPPGLs.** The data included correspond to 261 tumor-normal pairs from the CNIO and TCGA cohorts. **a** Overview of the number of variants, MSI score, and other clinical and genomic features of the whole series. Tumors have been ranked by TMB (calculated with category 5 variants). The legend in the bottom indicates the color code for each item. The filtering strategy for WES events is shown in Supplementary Fig. 2. **b** TMB and **c** MSI score (extracted with MANTIS) across PPGL tumors ($n = 256$). P-values were calculated with a two-sided Mann–Whitney–Wilcoxon (MWW) test and significant ones are shown in the figures. **d** Correlation between TMB and MSI score. Two-sided Pearson's correlation coefficient is shown. **e, f** Progression-free survival analysis of patients according to primary tumors' TMB and MSI score, respectively. Only primary tumors from non-metastatic and metastatic patients were analyzed. *Higher TMB* indicates tumors with values > than the third quartile ($n = 37$); *lower TMB* for the remainder cases ($n = 175$). *Higher MSI* score indicates cases with MSI score >0.15 ($n = 40$); *lower MSI* for the remainder cases ($n = 172$). Kaplan–Meier plots of time to progression (time between the first PPGL diagnosis and the first documented metastasis) are shown together with P-values calculated using a log-rank test. Median progression time (± standard error) of each group is depicted in the corresponding color. Patients

without evidence of metastases were censored at the date of the last follow-up. **g–i** TMB variation according to tumor volume (cm³) ($n = 28$), % Ki67 positive cells ($n = 16$) and *MKI67* mRNA expression ($n = 191$). Volumes and % Ki67 cells data, when available, were extracted from the pathological anatomy reports received with each specimen. *High MKI67 mRNA expression* indicates expression levels above the 3rd quartile value of the whole cohort and *low MKI67 mRNA expression* when levels were beneath the 3rd quartile value. **j** TMB variation according to the age of the patient at surgery ($n = 225$). For (**h**)–(**k**) a two-sided MWW was applied to test for differences, and, except for (**j**), only primary tumors (non-metastatic, aggressive and metastatic) were considered. **k** Frequency of *ATRX/TERT*-alterations within genomic subtypes. Two-sided Freeman–Halton test was used to test for differences between genomic subtypes. Metastatic tumors include primary tumors and metastases; if paired primary-metastasis is available, only one tumor per patient is represented. **l** TMB and **m** MSI score in *ATRX/TERT*-wild-type (WT) and in *ATRX/TERT*-altered tumors ($n = 256$). Two-sided MWW was used to test for differences. For all box-plots in this figure: the median value is marked, and Tukey whiskers are represented. See also Supplementary Fig. 3 and Supplementary Table 2. Source data are provided as a Source Data file.

**Table 1 | Univariate and multivariate logistic and Cox regression analysis of metastasis risk and TTP**

| Characteristics | Risk of metastasis development | | Time to progression | |
|---|---|---|---|---|
| | Univariate OR (95% CI), P | Multivariate OR (95% CI), P | Univariate HR (95% CI), P | Multivariate HR (95% CI), P |
| Female | 0.58 (0.30–1.08), 0.089 | 0.49 (0.15–1.54), 0.229 | 0.55 (0.31–0.97), 0.040 | 0.69 (0.36–1.34), 0.271 |
| Krebs cycle gene mutation | 8.24 (3.55–20.13), 1.5e-6 | 14.05 (3.42–64.82), 3.6e-4 | 2.19 (1.14–4.20), 0.018 | 1.54 (0.76–3.14), 0.232 |
| *ATRX /TERT*-alt | 9.10 (4.44–19.23), 3.1e-9 | 3.40 (0.78–15.41), 0.104 | 5.21 (2.90–9.36), 3.3e-8 | 1.83 (0.89–3.75), 0.102 |
| High MSI score | 68.25 (25.36–222.21), 9.7e-15 | 88.35 (23.29–448.89), 1.7e-9 | 11.44 (6.06–21.58), 5.5e-14 | 6.05 (2.89–12.65), 1.7e-6 |
| Mutational burden | 1.16 (1.11–1.23), 1.1e-9 | 1.14 (1.06–1.25), 0.001 | 1.08 (1.06–1.10), 1.5e-12 | 1.03 (1.00–1.05), 0.059 |

Multivariate analysis includes: sex, mutations in C1A cluster, alterations in *ATRX* and *TERT*, MSI score (as dichotomous variable established by MSI median value: 0 – MSI score < 0.15 and 1 – MSI score > 0.15) and TMB as covariates. Only data from primary tumors from non-metastatic and metastatic patients was included. To assess the degree of collinearity between our chosen explanatory variables, we ensured that the variance inflation factor (VIF) of each of them for the logistic regression was <2 (sex: 1.02; C1A cluster: 1.15; *ATRX/TERT* alterations: 1.5; MSI (>0.15): 1.14; TMB: 1.55).

independent indicator of prognosis (HR = 2.21, 95% CI = 1.11–4.79, $P = 0.045$). We also validated by immunohistochemistry (IHC) higher CDK1 protein levels in the nucleus of primary metastatic tumors (Fig. 4f), which were correlated with *CDK1* gene expression (Supplementary Fig. 5e).

In order to identify altered processes in metastatic tumors, and not only specific genes, we performed a GSEA analysis comparing metastatic *vs* non-metastatic primary tumors (Fig. 4g; Supplementary Fig. 6). This analysis revealed a positive normalized enrichment score (NES > 1.85; FDR < 0.05) not only in gene sets involved in cell cycle, DNA repair and p53 pathway, but also in those related to extracellular matrix (ECM) organization, translation/ ribosomes, ubiquitin/ proteasomes, and motility; this indicates that genes belonging to these categories are mainly upregulated in metastatic primary tumors. In contrast, gene networks associated with cilium formation, ion transport, Rho-GTPases, adhesion, nervous system (NS) development/ neuron projection and immune response displayed negative NES (NES < −1.85; FDR < 0.05), illustrating a downregulation of genes in these categories in metastatic primary tumors. Cilia loss and Rho pathway deregulation have already been reported in PPGLs[9,22,23], reinforcing the importance of these signaling networks in PPGL malignant transformation. Signaling cascades related to the immune system and deregulated in metastatic *vs* non-metastatic primary tumors (Fig. 4h; Supplementary Fig. 6o) included multiple sets involved in innate and adaptive immunity, cytokine secretion and inflammation that had not previously been associated with mPPGLs.

### Metastasis risk classifier for pheochromocytoma and paraganglioma

We evaluated the classification power of the potential markers of mPPGL described in this study: high MSI score, high TMB, gain of chr 5

and high *CDK1* expression. We also aimed to assess their power in combination with the already described markers of mPPGL (germline mutation in Krebs cycle genes, *MAML3*-fusion, *TERT*-alt, *ATRX*-mutation and high *MKI67* expression[5,8,15,19,24]). The area under the receiver operating characteristic curves (AUC under ROC) of each event individually and all combinations possible ($n = 511$) were computed. The AUC, 95% confidence interval (CI) and sensitivity/specificity per each classifier are in Supplementary Data 1. The best classifier is the one that takes into account *ATRX*-mutations, high MSI score, high *CDK1* expression and *MAML3*-fusions, showing a 100% sensitivity with the highest ability to predict mPPGL (AUC = 0.902, 95%CI = 0.855–0.948) (Supplementary Fig. 7).

### Functional enrichment analysis of mutated and CN-altered genes in metastatic pheochromocytoma and paraganglioma

Panther tool[25] was applied to detect significantly enriched gene sets among the list of genes with high impact consequence mutations in primary tumors from metastatic patients, and significantly differing focal SCNA between non-metastatic and metastatic primary tumors. The list of high impact mutated genes comprised 323 genes that grouped into 52 significantly enriched processes (fold-enrichment>1.8 and FDR < 0.05) classified into five different functional annotations: *circulatory system development/morphogenesis, ECM organization, cell adhesion and motility, actin cytoskeleton,* and *NS development and neuron projection* (Supplementary Fig. 8a). Focal SCNA significantly differing between non-metastatic and metastatic primary tumors and resulting in altered gene expression was present in 911 genes (see methods section). After applying a functional enrichment analysis using the 911 gene list, we recognized 120 significantly enriched gene sets (fold-enrichment>1.8 and FDR < 0.05) (Supplementary Fig. 9a).

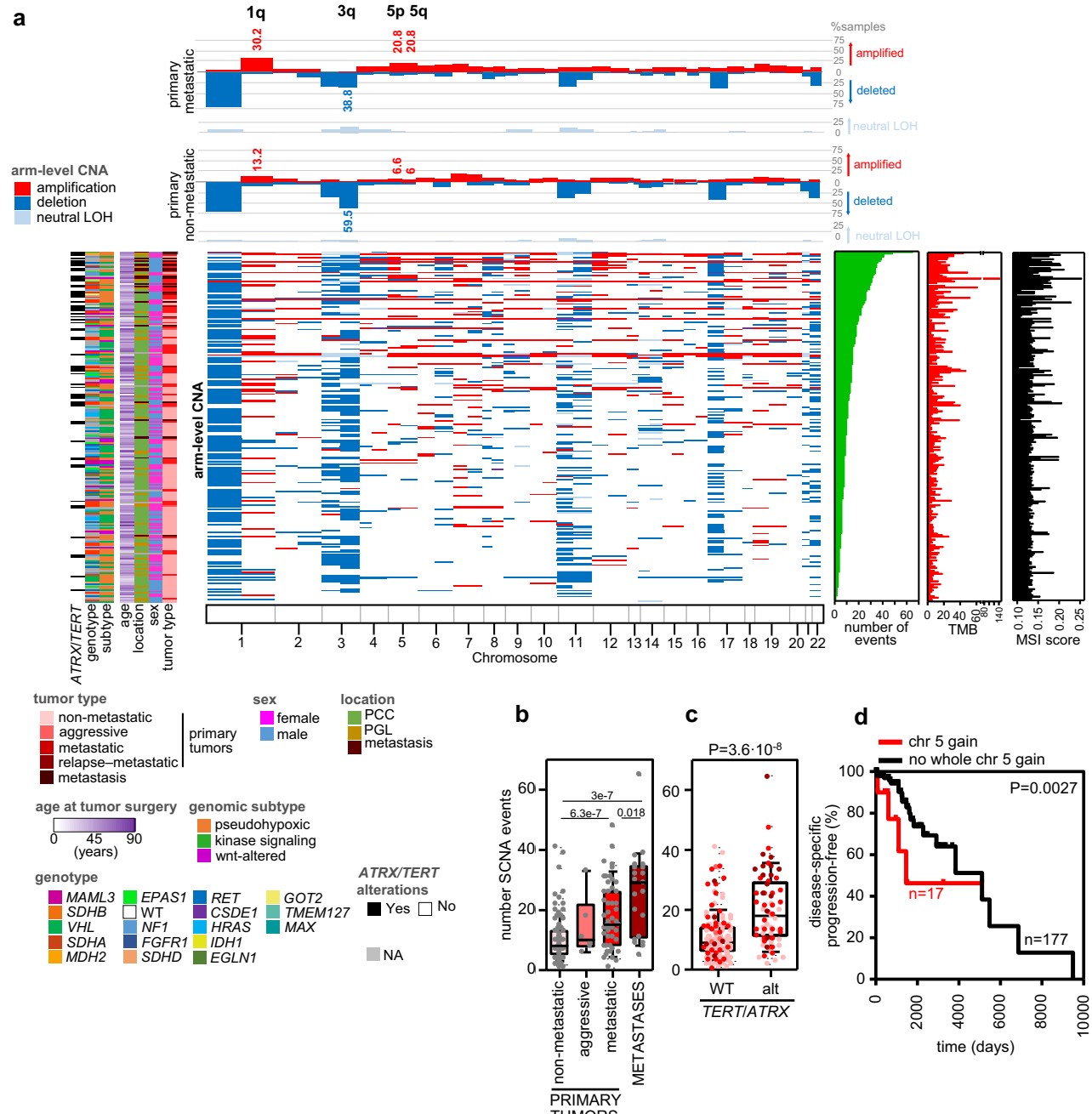

**Fig. 3 | Somatic copy number alterations (SCNA) profile in mPPGLs.** The data included is from 234 tumor-normal pairs from the CNIO and TCGA cohorts. **a** Overview of the arm-level SCNA, and clinical, molecular and technical features of the whole series. Tumors have been ranked by SCNA burden (number of SCNA events). The legend in the bottom indicates the color code for each item. Top panel: % of samples with arm-level CNA in metastatic and non-metastatic primary tumors. Significant (Fisher test; FDR < 0.1) regions between both groups are annotated. **b** SCNA burden (number of SCNA events) across PPGLs ($n = 232$). $P$ values shown were calculated with a two-sided MWW test. **c** SCNA in *ATRX/TERT*-wild-type (WT) and in *ATRX/TERT*-altered tumors ($n = 232$). Two-sided MWW was used to test for differences. **d** Progression-free survival analysis of patients according to the presence of whole chromosome 5 gains ($n = 17$) or no whole chromosome gains ($n = 177$) in primary tumors. Only primary tumors from non-metastatic and metastatic patients included. Kaplan-Meier plot of time to progression (time between the first PPGL diagnosis and the first documented metastasis) and the $P$-value calculated using a log-rank test is exposed. Patients without evidence of metastases were censored at the date of the last follow-up. For all box-plots in this figure: the median value is marked, and Tukey whiskers are represented. See also Supplementary Fig. 4. Source data are provided as a Source Data file.

Given the availability of RNA-Seq data, we executed GSEA with the 52 and 120 sets that resulted from the previous analysis in order to elucidate which categories were affected at the mRNA expression level. The analysis performed with the mutated gene sets (52) revealed differential expression (comparing metastatic *vs* non-metastatic primary tumors) of genes involved in five described functional annotations (Supplementary Fig. 8b), three of which were more relevant at the transcriptional level (>30% of sets of particular annotation with FDR < 0.05 in the GSEA analysis): *ECM organization*, *cell adhesion and motility*, and *NS development and neuron projection* (Fig. 5a). Notably, ECM and adhesion (Supplementary Fig. 6l, m) are terms classically linked to tumor progression and metastasis[26,27]. The weaker relative activity towards neuron differentiation (Supplementary Fig. 6k) may indicate a dedifferentiated neuronal-like phenotype in mPPGLs. This

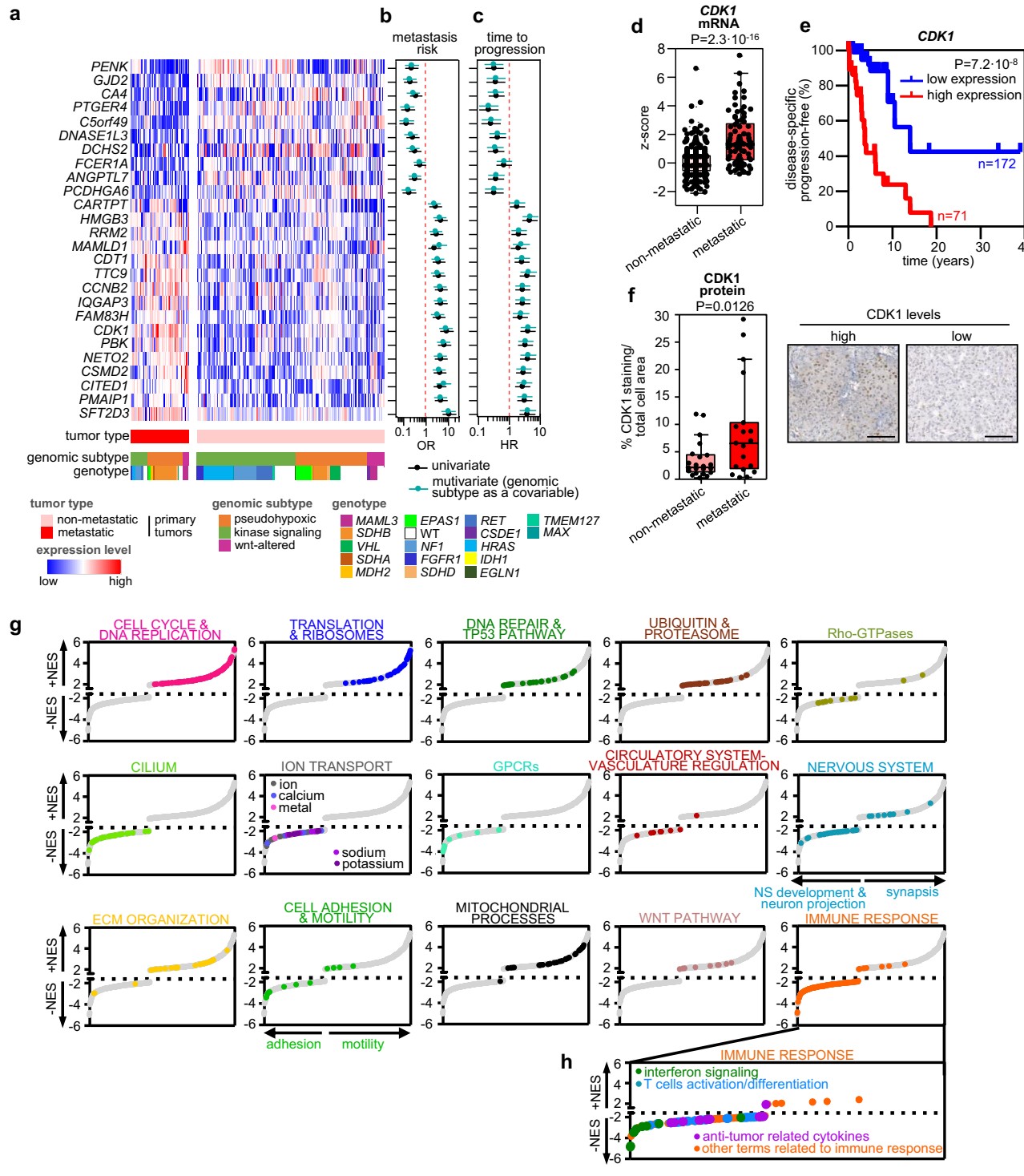

feature could be related to a neuroendocrine-to-mesenchymal transition associated with aggressive traits[28–30].

SCNA analysis identified gene sets enriched in mPPGLs, 98 exhibiting statistically significant differences between non-metastatic and metastatic primary tumors (Supplementary Fig. 9b; Fig. 5b). These could be classified into 5 different functional annotations: *translation and ribosomes, DNA repair and TP53 pathway, splicing, cell cycle and chromatin regulation* and *ubiquitin and proteasome*. Ribosome biogenesis has been reported to be essential for tumorigenesis and sufficient to drive malignant transformation[31]. Ribosome biogenesis, by preventing p53 ubiquitination and degradation[32,33], is also linked to the cell cycle[34]. Moreover, hyperactive rDNA transcription leads to DNA

damage and genome instability, activating DNA damage responses[31], that could also explain the higher TMB, MSI score and SCNA burden observed in mPPGLs (Figs. 2, 3).

Therefore, integration of transcriptomic and genomic data (mutations and SCNA), suggests causal mechanisms underlying 7 out of the 15 functional annotations deregulated at the transcriptional level (Fig. 4g).

## Immune landscape of metastatic pheochromocytoma and paraganglioma

Using a combination of gene expression analyses and IHC scoring with a set of immune markers, we characterized the immune

**Fig. 4 | mPPGL transcriptomic profile. a** Gene signature associated with mPPGL. mRNA expression levels of the 26 differential expressed selected genes, tumor behavior, genomic subtype and genotype are depicted in rows; primary tumors appear in columns (*n* = 230). Univariate (black) and multivariate (blue) logistic and Cox regression analysis of metastasis risk (**b**) and TTP (**c**), respectively. Gene expression was dichotomized as follows: for downregulated genes in mPPGLs, median expression was used as threshold (0 – below the median expression level; 1 – above the median expression level); for up-regulated genes in mPPGLs, high expression levels > than the 3rd quartile (0 – below the 3rd quartile threshold value; 1 – above the 3rd quartile threshold value). Multivariate analysis included as covariate genomic subtype. Only data from primary tumors from non-metastatic and metastatic patients were included. **d** Box plot of *CDK1* expression of primary tumors included in this study united to those from an independent cohort[20] (*n* = 417). Expression from both cohorts was z-score transformed (centered at the mean of non-metastatic group for each cohort). The P-value corresponds to a two-sided MWW test. **e** Progression-free survival analysis of patients according to *CDK1* expression. Kaplan−Meier plot of time to progression (time between the first PPGL

diagnosis and the first documented metastasis) is shown together with the *P*-value calculated using a log-rank test. High levels (above the 3rd quartile value of the whole cohort) are represented in red (*n* = 71) and low expression (below the 3rd quartile) in blue (*n* = 172). Patients without evidence of metastases were censored at the date of the last follow-up. **f** Representative images (right) and quantification (left) of CDK1 IHC staining in a subset of *n* = 41 PPGLs. Three PPGLs classified as aggressive were not included in this analysis. Scale bar in images = 100 μm. Unpaired two-sided t test was used to test differences between groups. **g** Signaling networks underlying mPPGLs. Plots show normalized enrichment score (NES) of significantly enriched gene sets (FDR < 0.05, GSEA) from MSigDB, grouped according to relevant biological processes. Each dot represents a gene set and is highlighted in the specific color for each process. **h** Close-up of gene sets gathered in *immune response* annotation and differentiation between *interferon signaling*, *T cells/activation differentiation* and *anti-tumor related cytokines*. For all box-plots in this figure: the median value is marked, and Tukey whiskers are represented. See also Supplementary Figs. 5, 6. Source data are provided as a Source Data file.

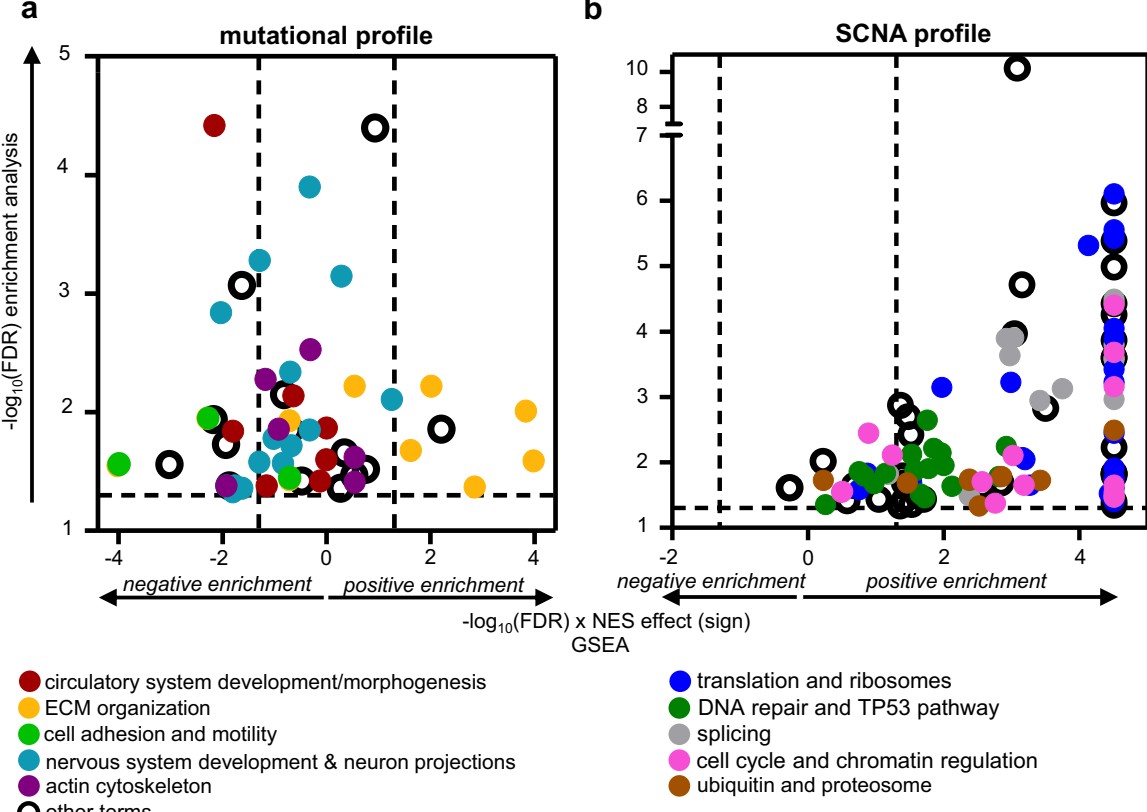

**Fig. 5 | Functional enrichment analysis of mutated (a) and CN-altered (b) genes in mPPGLs.** Functional enrichment analysis of mutated (**a**) and CN-altered (**b**) genes in mPPGLs. *y*-axis indicates the −log₁₀(FDR) of each gene set identified in the functional enrichment analysis performed using the list of 323 genes mutated in the metastatic primary tumors (**a**) or the 911 genes with differing SCNA between metastatic and non-metastatic primary tumors (**b**). Only those gene sets with fold-change enrichment (FCe) > 1.8 and FDR < 0.05 by Fisher's Exact test were

considered. *x*-axis indicates −log₁₀(FDR) × NES sign from the GSEA analysis performed using the gene sets recognized in the functional enrichment analysis and the RNA-Seq series of metastatic *versus* non-metastatic primary tumors. The gene sets related to each functional annotation are shown in different colors as depicted in the legend. Dashed lines indicate the cut-off values for significance. The extended version of the figure, in which each gene set name is annotated, is depicted in Supplementary Figs. 8, 9. Source data are provided as a Source Data file.

infiltration profile of each tumor. First, we used the data generated by Thorsson et al.[35] for the TCGA cohort and applied the *CRI-iAtlas/ ImmuneSubtypeClassifier* R package to classify the tumors from the CNIO series according to the available RNA-Seq data; from this, we established the distribution of the six immune subtypes described by Thorsson et al. (C1−C6) across our collection (with 0.38% tumors in C1, 48.66% in C3, 45.21% in C4, and 5.75% in C5). C3 (inflammatory) and C4 (lymphocyte depleted) were the predominant immune subtypes in PPGLs, but interestingly they were represented

differently according to tumor type; C3 was more abundant in non-metastatic tumors (52.3%), whereas C4 showed more prevalence among primary metastatic tumors and metastases (50.9 and 71.4%, respectively) (Fig. 6a). In addition, different proportions were also observed among genomic subtypes: C3 was more abundant in the pseudohypoxic subtype (72.2%), while C4 was more frequent in the kinase signaling subtype (67.6%). Of note, regardless of genomic subtype, more metastatic primary tumors and metastases were classified as C4 (*P* < 1 × 10⁻⁴), and those patients with tumors

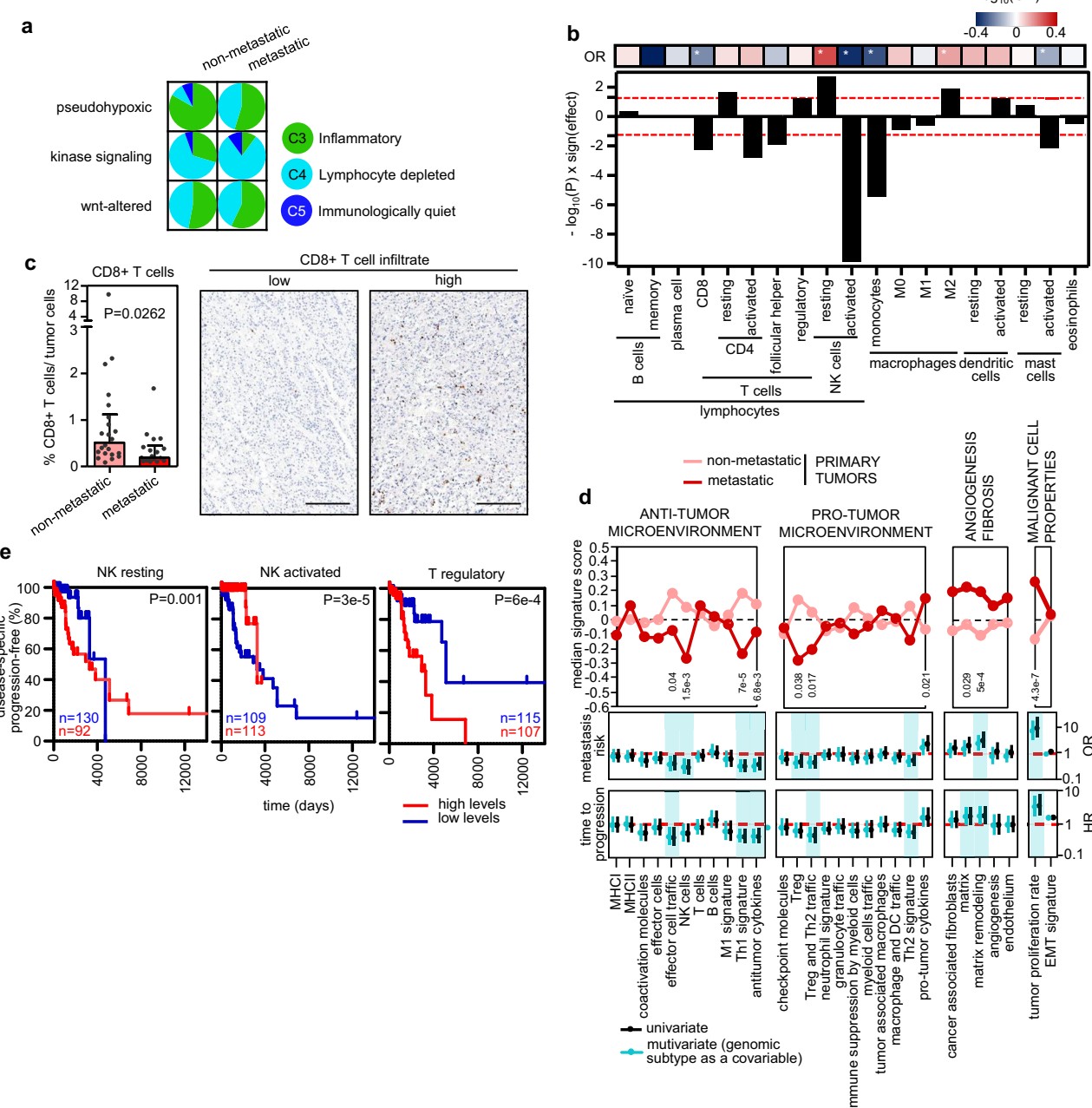

**Fig. 6 | Immune landscape in PPGLs. a** Classification of immune subtypes within tumor tissue types and genomic subtype. The proportion (%) of samples of each immune subtype per tumor type and genomic subtype is shown. Aggressive tumors were excluded from pie charts due to the low number of samples available. **b** Plot showing the −log₁₀(P) resulting from a two-sided MWW analysis to define differences between metastatic primary *versus* non-metastatic tumors in the different immune cell classes estimated with CIBERSORTx. The 'sign(effect)' indicates the direction of the fold-change between the proportions in both groups. Cell types with >85% of the samples with '0 s' were excluded from the analysis. Columns that surpass the red dashed line have *P* < 0.05. The top row summarizes univariate logistic regression analysis comparing immune cell proportions and metastatic risk. The color of the cell is relative to the −log₁₀(OR), and * indicates *P* < 0.05. **c** Percentage of CD8+ T cells infiltrated among tumor cells detected by immunohistochemistry (left) in a subset of *n* = 39 PPGLs with different clinical behavior. Three PPGLs classified as aggressive were not included in this analysis, and for two cases IHC were not assessable. Median ± IQR is shown. Two-sided MWW was applied to test for differences. Representative images (right) of CD8 IHC. Scale bar in images = 200 μm. **d** Top panel: median enrichment score of the different Fges in

metastatic and non-metastatic primary tumors. Two-sided MWW was applied to test for differences between metastatic (*n* = 55) and non-metastatic (*n* = 176) primary tumors; significant *P* values are shown. Univariate (black) and multivariate (blue) logistic and Cox regression analysis of metastasis risk (middle panel) and TTP (bottom panel), respectively. Enrichment scores were used as a continuous variable. Multivariate analysis included genomic subtype as covariate. Only data from primary tumors from non-metastatic and metastatic patients were included. Significant associations in Fges scores after multivariate analysis are shaded in blue. **e** Kaplan–Meier plots of time to progression in patients according to different immune cell type levels found in primary tumors. Only primary tumors from non-metastatic and metastatic patients included. High levels (above the median level of the whole group) are represented in red (*n* = 92, *n* = 113, and *n* = 107, respectively for NK resting, NK activated and T regulatory) and low expression (below the median level) in blue (*n* = 130, *n* = 109 and *n* = 115, respectively for NK resting, NK activated and T regulatory). *P*-values shown inside the plots were calculated using a log-rank test. Patients without evidence of metastases were censored at the date of the last follow-up. See Supplementary Fig. 10 for the extended version. Source data are provided as a Source Data file.

categorized as C4 seemed to have a shorter TTP than those classified as C3 (Supplementary Fig. 10a).

Next, we used the immune score, extracted by ESTIMATE[36], to evaluate the presence of immune cells in the tumors using RNA-Seq data. Although we did not observe different infiltration levels between the different tumor types (Supplementary Fig. 10b), we discovered substantial variations in the proportion of immune cells extracted by deconvolution of the RNA-Seq data using CIBERSORTx[37] (Fig. 6b). We observed a significantly higher number of resting and lower number of activated CD4+ T cells, lower proportion of CD8+ T cells, and a higher number of resting and lower quantity of activated NK cells in metastatic primaries compared to non-metastatic tumors. The lower levels of CD8+ T cells in metastatic tumors was validated by IHC (Fig. 6c). The suppression of activated T cells is supported by the GSEA analysis with 24 terms related to activation/differentiation/proliferation identified among the negative NES (Fig. 4h; Supplementary Fig. 6o, blue arrows). This notion was reinforced by the differences between non-metastatic and metastatic primary tumors found in the functional gene expression signatures (Fges) enrichment scores, calculated by GSVA as in Bagaev et al.[38], related to effector cell traffic and NK cells (Fig. 6d, top panel). We also noted a higher representation of macrophages M2-like, known by their anti-inflammatory and pro-tumoral role[39], in metastatic primary tumors. This is in agreement with the observation that a higher proportion of metastatic tumors were classified into C4 subtype, characterized by high M2-like macrophage response[35].

Several gene sets ($n = 9$) related to interferon signaling also had negative NES in the GSEA analysis (Fig. 4h; Supplementary Fig. 6o, green arrows), and a significantly lower *Th1 signature*- and *anti-tumor cytokines*-, and higher *pro-tumor cytokines*-Fges scores were found in metastatic primary tumors (Fig. 6d, top panel), both supporting the immunosuppressive microenvironment encountered in the metastatic primary tumors. These differences increased when metastases were incorporated into the analysis (Supplementary Fig. 10c). Other cytokines with anti-tumor immunity properties, such as TNF, IL6, IL2, and IL1[40], were also identified by the GSEA analysis (Fig. 4h; Supplementary Fig. 6o, purple arrows).

Interestingly, lower *NK cells*-, *effector cell traffic*-, *Th1 signature*- and *antitumor cytokines*-Fges scores were associated with risk of metastasis development (multivariate analysis adjusting for genomic subtype; Fig. 6d, middle panel), suggesting that TME is independent of genotype.

The latter Fges scores were also associated with TTP (Fig. 6d, bottom panel). We also found that some infiltrating immune cell types estimated with CIBERSORTx were significantly associated with TTP; high levels of resting and low levels of activated NK cells were associated with shorter TTP (Fig. 6e; Supplementary Fig. 10d). Similarly, high levels of regulatory T cells (Tregs) were also associated with shorter TTP.

## Immunogenomics as a theranostic tool in the immunotherapy contexture of metastatic pheochromocytoma and paraganglioma

Bagaev et al. recently proposed four Fges-based pan-cancer microenvironment subtypes that correlate with patient response to immunotherapy in multiple cancers[38]. To classify our tumors according to their TME, we applied k-means clustering using the 29 Fges scores. We defined four TME subtypes in our cohort (Fig. 7a); among these, the immune-enriched non-fibrotic was the least represented (14.3% of the cases), and the subtype with fewer metastatic tumors (Supplementary Fig. 11a) and with better prognosis (Fig. 7b). TME subtypes were strongly correlated with the ImmuneScore and StromalScore, obtained from ESTIMATE[36], and predicted levels of infiltrating immune and stromal cells, respectively ($P = 4 \times 10^{-38}$ and $5.4 \times 10^{-44}$, Krustal–Wallis test) (Fig. 7a). We also observed a trend towards TME subtypes being associated to neoantigen load and TMB

(Supplementary Fig. 11b, c). TME subtype proportions also differed between immune subtypes (Supplementary Fig. 11d) and PPGL genomic subtypes (Supplementary Fig. 11e). The pseudohypoxic cluster was mainly enriched in the fibrotic subtype, characterized by high vascularity and an immunosuppressive phenotype[38] as recently reported by Celada et al.[41]; in contrast, the kinase signaling cluster was highly represented in the immune-enriched non-fibrotic subtype, reflecting the influence of the genomic subtypes on TME. These differences were also evident when comparing Fges scores between genomic subtypes (Supplementary Fig. 11f). We observed higher representation of a Th1 signature and checkpoint molecules in the kinase signaling tumors, and the pseudohypoxic tumors were characterized by higher signature Fges scores related to angiogenesis and fibrosis (cancer-associated fibroblasts, matrix remodeling, endothelium), as already reported[42,43].

In order to further explore the immunophenotyping of patients with mPPGL, we examined the expression of key immunomodulators (IMs) targeted in clinical oncology by several IM agonists and antagonists[44]. Gene expression of IMs (Fig. 7c; Supplementary Fig. 12a) varied between metastatic and non-metastatic tumors and across genomic subtypes, possibly suggestive of their role in delineating the immune infiltration of the tumors. Genes with the greatest differences ($P < 0.01$) between metastatic and non-metastatic tumors were *CD274*, *CD276*, *TGFB1*, *VEGFA*, *CD40* and *TLR4*. *CD274*, which encodes for programmed death-ligand 1 (PD-L1), showed the lowest levels in pseudohypoxic tumors and the highest in Wnt-altered subtype, including several metastatic *MAML3*-tumors. PD-L1 IHC in an independent series of $n = 44$ PPGLs (Supplementary Table 3) confirmed that 34% (15/44) of PPGLs, 7 harboring a *MAML3*-fusion, exhibit PD-L1 staining in tumor cell membranes. These represented 100% of *MAML3*-tumors included in the series and, with the exception of one *SDHB* tumor, were the only tumors exhibiting highly positive PD-L1 staining (Fig. 7d); no association was observed according to clinical behavior (Supplementary Fig. 12b). A higher neoantigen load was also observed in *MAML3*-tumors compared to all other tumors (MWW test: $P = 0.0067$; Fig. 7e).

Given the differences observed in the aforementioned key immune players, further investigation is needed in order to elucidate the differences in terms of immune infiltrates between the tumors classified in the different genomic subtypes. We observed differences between molecular clusters according to Fge scores, immune and TME classifications (Supplementary Fig. 10c, d; Fig. 6a), and, as anticipated, we also noticed differences according to relative levels of some immune cell types among genomic subtypes (Supplementary Fig. 13, top). One difference was the greater infiltration of CD8+ T cells in *MAML3*-tumors (Fig. 7f). On the other hand, we show that metastatic tumors present a similar pattern in the proportion of major classes of immune cells across the different genomic subtypes (Supplementary Fig. 13, bottom), reinforcing the notion that some immune cell populations may be partially linked to the molecular genomic subtype, but others to the tumor behavior.

## Discussion

The comprehensive genomic analysis of mPPGL described here highlights not only the heterogeneous landscape of secondary genomic alterations, but also the presence of shared TME features within genomic subtypes and/or according to tumor behavior.

In this study, we reveal that TMB, MSI status and SCNA burden of PPGLs increase with disease progression, suggesting their use as markers of metastatic risk at diagnosis of a primary tumor. We confirm differential TMB in PPGLs as previously reported[5,9], and report the association between changes in MSI status and clinical behavior of PPGLs. Although we did not identify any PPGL tumor surpassing the cutoff value used for MSI-high classification in tumors with high TMB[10,13], we propose that MSI thresholds may not be applied similarly

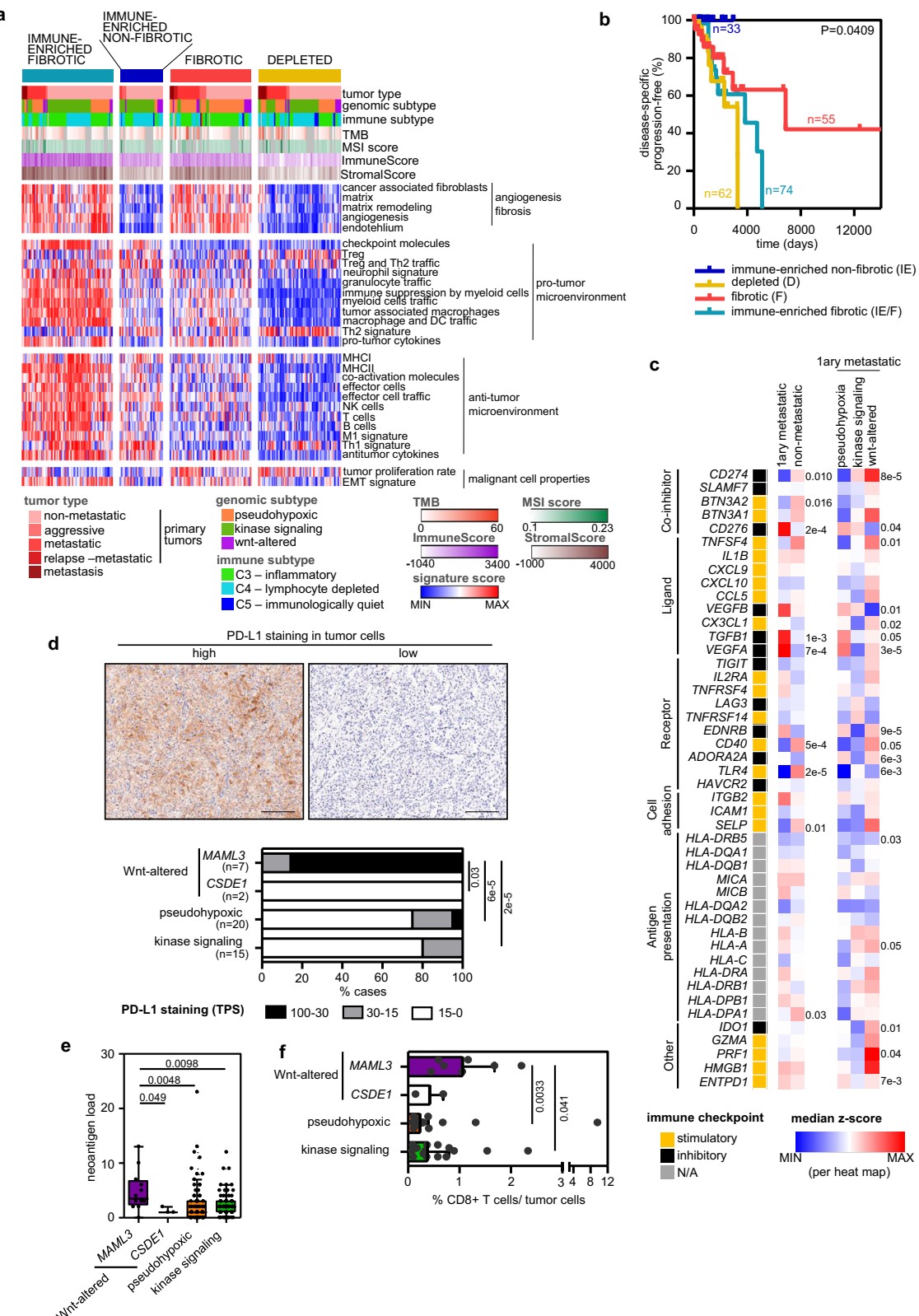

to tumors with lower TMB, such as for PPGL or acute myeloid leukemia[45,46]. Frequent gain of chromosomal material had previously been observed in mPPGL[47], but here we have linked an increased SCNA burden to mPPGL, mimicking observations in other tumors[48]. Importantly, TMB, MSI status and SCNA burden were associated with *ATRX/TERT* alterations, reinforcing the notion that alterations in immortalization-related genes contribute to genomic instability[49]. In

fact, different copy number signatures could underlie this heterogeneous genomic instability, as reported for ovarian cancer[50].

*ATRX/TERT* alterations are validated as the only recurrent secondary prognostic-associated events identified in PPGLs, accumulating not only in pseudohypoxic tumors[8,19,51,52], but also in Wnt-altered tumors. These findings bolster the concept that alterations contributing to immortalization mechanisms are those that confer a poor

**Fig. 7 | Immunogenomics as a theranostic tool in the immunotherapy contexture. a** Heatmap of the 267 PPGL tumors profiled by RNA-Seq and classified into the four distinct TME subtypes described Bagaev et al. based on unsupervised k-means clustering of the 29 Fge enrichment scores. Genomic and clinical features are exhibited as depicted in the legend. **b** Kaplan–Meier plot of time to progression in patients according to the primary tumor TME subtype ($n = 33$ for IE, $n = 55$ for F, $n = 74$ for IE/F and $n = 62$ for D). Only primary tumors from non-metastatic and metastatic patients included. *P*-value was calculated using a log-rank test. Patients without evidence of metastases were censored at the date of the last follow-up. **c** Expression (median normalized z-score expression) of main immunoregulators (IMs) according to the clinical behavior and genomic subtype. Two-sided MWW and two-sided Krustal-Wallis tests were applied to test for differences between metastatic ($n = 55$) and non-metastatic ($n = 176$) primary tumors and between the different genomic subtypes (pseudohypoxic, $n = 33$; kinase signaling, $n = 16$; Wnt-altered, $n = 6$) within metastatic primary tumors, respectively; both tests were applied on the normalized data. Significant *P*-values are shown. **d** Representative images (top) and quantification (bottom) of PD-L1 IHC staining in a subset of $n = 44$ PPGLs. Quantification is represented for each genomic subtype. Two-sided Freeman–Halton test was used to test for differences between groups. Only *MAML3* class exhibited statistically significant differences compared to the other groups. Scale bar in images = 150 μm. **e** Neoantigen load across genomic subtypes in primary tumors ($n = 170$). Only tumors with WES and RNA-Seq data available were included in the analysis; WT Wwnt-altered tumors have not been represented. The median value is marked and Tukey whiskers are represented. *P*-values shown in the figure correspond to two-sided MWW test. Only *MAML3* class exhibited statistically significant differences compared to the other groups. **f** Percentages of CD8+ T cells infiltrated among the tumor cells detected by immunohistochemistry in a subset of $n = 42$ PPGLs with different genomic subtypes. For two cases, IHC was not assessable. Median ± IQR is shown. MWW was applied to test for differences between groups. See also Supplementary Figs. 11–13. Source data are provided as a Source Data file.

prognosis for patients with PPGL. Except for *ATRX* mutations, *TERT* alterations and chromosome 5 gains, no other genetic alterations could be identified as metastasis priming events. Therefore, and even though PPGL tumor formation is clearly driven by germline or somatic mutations, our data suggest that the molecular switches involved in metastatic progression are likely variable and may involve non-genetic events. However, of course, future efforts should also be directed towards performing WGS to capture potential alterations in non-coding regions or large rearrangements[53].

At the transcriptional level, we identified elevated *CDK1* as a prognostic factor that, independently of genomic subtype, adds to the higher TMB, higher MSI score and chromosome 5 gains just described herein. Moreover, given that CDK1 is indispensable for driving the cell cycle[54], and because the cell cycle is one of the major deregulated signaling networks in mPPGLs (described herein and recently by Zethoven et al.[55]), there may be potential for developing CDK1-specific inhibitors for treatment[56].

Given what has been reported to date[57], it will be difficult to identify a single molecular marker capable of accurately predicting the metastatic behavior of these tumors. Therefore, it is of particular interest that our study describes a classifier that takes into account only 4 events (*ATRX*-mutations, high MSI score, high *CDK1* expression and *MAML3*-fusions) and achieves a sensitivity of 100%. Although further validation will be necessary, we have defined a classifier that could identify all patients at risk of metastasis at the time of diagnosis of the primary tumor.

The immune contexture is being evaluated as a prognostic tool in many tumor types[58], but little has been reported in PPGLs[41,55,59,60]. Here, we identified a high number of downregulated gene sets related to the immune response. By interrogating the immune content, we also showed a complex and heterogeneous ecosystem at the TME level, characterized by the presence of CD8+ T cells, NK cells and a Th1 profile; this was associated with a good clinical outcome, as expected, due to their anti-tumor immune characteristics[61]. Additionally, we identified an association between high levels of macrophages M2 and Tregs, both cell types known by their anti-inflammatory and pro-tumoral role[39,62], and metastatic behavior and poorer prognosis. The excessive catecholamine levels present in metastatic tumors could contribute to the establishment of the immunosuppressive TME[63], by diminishing T and NK cell activation[64] and by driving the M1- to M2-like macrophage polarization[65]. Our data demonstrate the potential usefulness of including immune parameters as prognostic classification tools for PPGL.

Cancer immunotherapy is in the spotlight for the treatment of numerous cancers and the TME is known to be important for tumor progression and anti-cancer therapeutic responses[66]. Therefore, it is possible that classification of PPGLs according to their TME subtype[38] could help identify patients who might benefit from immunotherapeutic agents. We established that tumors with immune-enriched non-fibrotic subtype, showing an immune-active TME, exhibited the best prognosis and included mainly non-mPPGLs. In melanoma, bladder and gastric cancer, responders to immune checkpoint inhibitor have the highest proportion of immune-enriched non-fibrotic subtype[38]. However, what we observe in mPPGL is discouraging. Specifically, the fibrotic subtype, which in our cohort was enriched in mPPGLs, seems to be the least responsive to the aforementioned therapies[38].

Strategies to individualize immunotherapy based on patient- and tumor-specific characteristics are a current focus of the scientific community[67]. With this in mind, we identified PD-L1 positivity, higher TMB and neoantigen load, and CD8+ T cell infiltration in *MAML3*-tumors compared to other PPGLs. This is noteworthy because all these traits are associated with improved responses to PD1 axis blockade in diverse tumors[68,69]; this suggests that PD-1/PD-L1 inhibitors may offer a potential treatment for patients with *MAML3*-tumors. Pembrolizumab, a PD-1 inhibitor displayed modest therapeutic efficacy in a clinical trial of mPPGLs[70]; although no association with genotype was observed, *MAML3* status was not considered, so it remains possible that this could be used to stratify patients. This concept is supported by findings that gene fusions are a source of immunogenic neo-antigens that can mediate responses to immunotherapy, even with a low TMB and minimal immune infiltration[71]. Of relevance, it is known that WNT/β-catenin signaling, upregulated in *MAML3*-tumors[5], correlates with immune exclusion[72]. The latter is caused by an increased PD-L1 transcription after MYC or β-catenin binding to its promoter[73,74]. Therefore, these observations suggest that WNT inhibitors may synergize with PD1 axis blockade in *MAML3*-tumors.

We could speculate that *CD40* and *TLR4* low expression identified in mPPGLs could pave the way to propose other approaches to expand the range of targetable immune checkpoints, as in situ vaccination with double CD40-TLR4 stimulation[75] to turn "cold tumors" into "hot tumors" and restore PD-1 sensitivity. Notably, CD40 and/or various TLR agonists, currently in clinical trials, have recently been tested in a PPGL mouse model with promising results[76].

The notion that the tumor immune portrait is orchestrated by the genomic subtype and that is modulated by the tumor behavior is supported by reported examples that show (i) a link between tumor genotype and immunological architecture, and (ii) the dynamic state of TME during tumor progression[77].

In summary, this study serves as a valuable resource for future investigations in the field of PPGL genomics due to the large series of tumors with WES and RNA-Seq data associated with pathological, IHC and curated clinical information. Moreover, the present report contributes with significant data to advance the genomic and immune understanding of mPPGLs, and thereby potentially increase the therapeutic strategies needed for a bench-to-bedside clinical success.

## Methods

### Sample cohort and clinical data

All PPGL tumor tissues and patient's germline samples were obtained from patients with informed written consent for this purpose, in accordance with institutional ethical-approved protocols. The study was conducted in accordance with the Declaration of Helsinki, and the protocol was approved by the following Ethics Committees: Hospital Universitario 12 de Octubre (15/024), Madrid, Spain; Universitäts Spital Zurich (2017-00771), Zurich, Switzerland; Klinikum der Universität (379-10), Munich, Germany; University of Würzburg (88/11), Würzburg, Germany; Azienda Ospedaliera Universitaria Careggi (Prot. N. 2011/0020149), Florence, Italy; Berlin Chamber of Physicians (Eth-S-R/14), Berlin, Germany; Radboud University Medical Center (9803-0060), Nijmegen, The Netherlands; TU Dresden (EK210052017 and EK189062010), Dresden, Germany. Clinical characteristics are provided in Supplementary Tables 1 and 3. Demographic and clinical data included sex, age at initial pathologic diagnosis, age at tumor surgery, tumor type (PGL, PCC, multiple), number of follow-up days for non-metastatic patients/ event free days until metastasis for patients with metastases, and location of metastasis. Histological annotated data, when available, included tumor volume, presence of capsular/ adipose tissue/ vascular invasion, and Ki67 index. Metastatic disease was defined as the occurrence of distant metastases in anatomical locations where chromaffin tissue is normally absent, as per WHO definition[78]. Aggressive disease was established when there was capsular or adipose tissue invasion, and vascular infiltration or evidences of multiple recurrences, without certainty of metastatic disease. Tumors that appear at the same place as the original primary after a period of time were tagged as "relapses".

Tumor tissues and germline samples were collected from 73 patients for WES analysis. After discarding samples that failed different quality control steps, a final cohort of 87 tumors from 61 patients, sequenced using 'SureSelect All Exon + COSMIC v6' exome probes from Agilent, were available for analysis (Supplementary Fig. 1a). From 165 patients initially available for the RNA-Seq study, a final series of 114 samples from 105 patients profiled either using Illumina TruSeq or QuantSeq 3'mRNA-Seq (Lexogen) and that passed our pre-analysis criteria, were included in the study (Supplementary Fig. 1b). To note, we profiled 45 tumor samples with both WES and RNA-Seq. Moreover, we included 20 patients with more than one tumor sequenced by WES (range, 2–7), and 9 patients with more than one tumor profiled by RNA-Seq (range 2–3). To increase the power of our analyses and overcome the lack of non-metastatic patients included in our WES series, we joined our WES and RNA-Seq data with those from the TCGA. After our in-house quality control of the latter series, we gathered a total of 261 paired germline-tumor analyzed by WES and 264 tumors profiled by RNA-Seq (Supplementary Fig. 1). Clinical data from patients included in the TCGA series was extracted from ref. [5].

The WES series was composed of 62 primary tumors and 5 relapses from metastatic patients, 25 metastases, 8 primary tumors from patients with aggressive disease and 158 primary tumors from non-metastatic patients (Supplementary Fig. 1a). The RNA-Seq series was composed of 55 primary tumors and 4 relapses from metastatic patients, 16 metastases, 15 tumors from patients with aggressive disease and 176 tumors from non-metastatic patients (Supplementary Fig. 1b).

The resulting series is representative of the main PPGL genotypes and equitable in number of samples from non-metastatic and metastatic patients. In the CNIO cohort, germline and somatic mutations in driver genes had previously been characterized using a NGS AmpliSeq Custom DNA Panel (Illumina, San Diego, CA, USA) as specified in[79]. MAML3 fusion positive tumors were detected either (1) on cDNA of suspected candidates using the following primers for the upstream UBTF portion of the fusion transcript: 5'-GGAGCAGCAAAAGCAGTACA-3' and 5'-CCCAAACCCCAAATCCAG-3'; and 5' TCTCCATTAAGT

GGTGATCGAG 3' for the downstream MAML3 portion, or (2) by fluorescence in situ hybridization (FISH) for MALM3 gene rearrangement analysis.

### Nucleic acid extraction and quality control

Tumor specimens were collected either in formalin-fixed paraffin-embedded (FFPE) or fresh-frozen (FF), and tumor selection (>60% cancer cells) was performed by a pathologist (E. Caleiras) on hematoxylin and eosin-stained slides. DNA and RNA were isolated using commercially available kits according to manufacturer's specifications. DNA from FFPE tissues was obtained with truXTRAC FFPE DNA (Covaris, 520136) and from FF tumors from DNeasy Blood and Tissue Kit (Qiagen, 69506). RNA from FFPE was extracted using Maxwell RSC RNA FFPE Kit (Promega, AS1440) and from FF with TRIzol Reagent (Invitrogen, 15596026). Germline DNA was extracted either from peripheral blood mononuclear cells using Maxwell RSC Whole Blood DNA Kit (Promega, AS1520) or from adjacent normal tissue using truXTRAC FFPE DNA (Covaris, 520136). DNA and RNA was quantified using Quantus Fluorometer (Promega) and integrity examined using a High Sensitivity DNA Kit (Agilent, 5067-4626) or a RNA 6000 Nano Kit (Agilent, 5067-1511) in a 2100 Bioanalyzer Instrument (Agilent). Samples with low quality DNA were dismissed.

### WES

**Library preparation and sequencing.** Whole exome sequencing (WES) libraries were prepared with the SureSelectXT Human All Exon V6+ COSMIC target enrichment system (Agilent, 5190–9307) following the manufacturer's instructions. 200 ng of genomic DNA from FFPE tumoral samples and 1 μg from FF or germline samples were used as starting material. SureSelectXT HS (Agilent, G9704A) and SureSelectXT2 (Agilent, G9621A) reagent kits were used for FFPE derived and good quality samples, respectively. DNAs were fragmented on a Covaris S2 sharing instrument based on initial DNA sample's integrity. Fragmented DNA samples were processed through subsequent enzymatic treatments of end-repair, dA-tailing, and ligation to Illumina's adapters. Adapter-ligated libraries were PCR amplified with Illumina PE primers and subsequently hybridized to SureSelect Oligo Capture Library Mix. Biotinylated hybrids were captured and the enriched fraction eluted. Libraries were completed by limited-cycle PCR with Illumina TruSeq primers and KAPA HiFi HotStart DNA pol (Roche KK2501). Sample-specific libraries were combined into equimolarly balanced pools and applied to Illumina flow cells for cluster generation. Sequencing was performed on either the Illumina HiSeq2500 or NovaSeq6000 in a paired-end 100 bp reads mode to a median target coverage of ~200× and ~100× for tumors and germline, respectively.

### WES analysis

Exome sequencing data was processed and aligned to the human reference genome GRCh37 using Burrows-Wheeler Aligner (BWA)[80]. For single-nucleotide variant (SNV) and insertions/deletions (INDELs) identification, the Genome Analysis Toolkit GATK[81] was used. Germline and somatic variants with higher allele frequency were detected with Haplotype Caller[82] and additional somatic variants were detected with MuTect[83] that was also incorporated to the pipeline to identify somatic variants with low allele fraction values. Variants were annotated using Ensembl Variant Effect Predictor tool (VEP)[84] v90. For the analysis employing TCGA DNA sequencing data, whole exome mapped files (BAM files), aligned against GRCh38 version of human reference genome were downloaded from GDC Data Portal, NIH (https://portal.gdc.cancer.gov/).

Tumor purity was estimated using FACETS algorithm[85], particularly the R package *cnv-facets* that allows the execution of all the necessary steps in a single command line call. GRCh37 and GRCh38 versions of human reference genome were used for our series and

TCGA series, respectively. Only samples with tumor cellularity >12% (median, 70.5%; range 13–92.3%) were included for analysis.

Microsatellite instability (MSI) was calculated using the MANTIS algorithm[86]. Normal-tumor paired, aligned and processed BAM files, were used to calculate the differences of microsatellite distributions at somatic level.

SNVs and indels were manually curated following the workflow shown in Supplementary Fig 2a. Briefly, events that fell into any of these categories were removed:

- Events annotated as: "intron_variant", "downstream_gene_variant", "3_prime_UTR_variant", "upstream_gene_variant", "NMD_transcript_variant", "non_coding_transcript_variant", "5_prime_UTR_variant", "mature_miRNA_variant".
- Variants with minor allele frequency > 0.1 in gnomAD or/and 1000genomes ('germline-like').
- Events visualized using IGV[87] in other patient's tumor WES and not appearing in their callings, or which appear in germlines of other patients were considered 'artefacts'.
- Events with variant allele fraction <0.1.

Variant impact consequence classification of the remaining events was applied as indicated in Supplementary Table 4.

For TCGA, WES data were extracted from 8 independent pipelines and the annotated files were obtained from UCSC Xena (accessed on Dec 3rd 2019). Data sets included SNPs and small INDELs detected after performing variant calling with MuSE, MuTect2, SomaticSniper and VarScan2 Variant Aggregation and Masking pipelines, as well as the calling generated by BCGSC (British Columbia Genome Sciences Centre, BCGSC pipeline), BCM (Baylor College of Medicine Human Genome Sequencing Center, Baylor pipeline), Broad (Broad Institute Genome Sequencing Center, MuTect pipeline) and UCSC (University of California Santa Cruz, RADIA pipeline). We generated a 'consensus' variant set including all those events called by at least two of the eight pipeline lists. The original TCGA coordinates in GRCh38 were converted to GRCh37 using the online tool LiftOver from UCSC (https://genome.ucsc.edu/cgi-bin/hgLiftOver). VEP[84] v94 was used to predict the functional effects of the variants. SNVs and INDELs were manually curated following the same workflow used for CNIO WES data, shown in Supplementary Fig. 2b, except for the artefact filtering step.

Somatic copy number (CN) detection was performed with FACETS algorithm[85], following the same analysis previously mentioned for purity estimation. Good fit of CN detection was evaluated with the diagnostic plot generated by the pipeline, and samples with poor fit were discarded from the analysis. For specific samples, regions with total CN value higher than 8, and regions with log-ratio values between −0.1 and 0.1 were discarded, except for regions annotated as neutral LOH. Total number of CN segments (defined as SCNA events) was calculated for each sample, and genes in each region were annotated using an in house R script based on biomaRt package[88].

## Identification of TERT alterations

*TERT* overexpression, *TERT* amplification, *TERT* promoter (*TERT*p) mutation and hypermethylation were taken into account in our analysis. When RNA-Seq data was not available, *TERT* overexpression was detected by four *TERT* gene expression probes included in a TruSeq Targeted custom panel (Illumina) that contained other 227 probes for 187 genes; only samples with an average value ≥4 counts were considered as positive for *TERT* expression. *TERT*p recurrent mutations (chr5:1295228C >T and chr5:1295250C >T) were studied using the 16S Metagenomic Sequencing Library Preparation (Illumina) protocol; only the C228T mutation was detected in our series. Amplicons of 151 bp for NGS were obtained from 50 ng of DNA tumor samples after two PCRs; Multiplex QIAGEN 2× Master Mix was used for the first PCR and EasyTaq DNA polymerase for the second. Mutations detected by NGS were validated using Sanger sequencing using forward and

Reverse primers (5′-TCGTCGGCAGCGTCAGATGTGTATAAGAGACAG CAGCGCTGCCTGAAACTCG-3′ and 5´- GTCTCGTGGGCTCGGAGAT GTGTATAAGAGACAGGTCCTGCCCCTTCACCTTC-3′).

*TERT*p hypermethylation was studied using 100 ng of bisulfite modified DNA (EZ-96 DNA methylation kit, Zymo research). Primers were chosen according to Castelo-Branco et al.[89], obtaining four amplicons covering the *TERT*p region CpG islands, as known as *TERT* hypermethylated oncological region (THOR). A total number of 39 probes from the THOR region were studied, sharing 29 probes with the ones included by Castelo-Branco et al. Methylation status was determined by focusing mainly in the UTSS region (CpG probes: chr5_1295586_C; chr5_1295590_C; chr5_1295593_C; chr5_1295605_C; chr5_1295618_C). Only samples with a mean UTSS value ≥14.1 were considered as hypermethylated.

High-level amplifications of *TERT*, identified by using GISTIC 2.0 tool[90] with a score = 2, were considered after data processing as described in Monteagudo, et al.[79].

TERT overexpression and TERTp hypermethylation in TCGA samples were established as described by Barthel et al.[91].

## Impact of sequencing depth coverage—downsampling

To evaluate the impact of the sequencing depth on variant calling sensitivity between CNIO (median, 91.9×) and TCGA (median, 53.5x) WES series, we selected 10 random samples from the CNIO series, and BAM files were downsampled ≈50× average coverage (similar to the mean depth of the TCGA cohort). We then reran the same somatic variant calling pipeline to compare it to the variant calling performed with the original depth (Supplementary Fig. 14a). Decreasing coverage resulted in a median diminution in sensitivity of only 11.8% in the number of variants detected (Supplementary Fig. 14b), and a good agreement in tumor purity estimation (Supplementary Fig. 14c).

## RNA-Seq

**RNA-Seq Library preparation and sequencing.** RNA library preparation of samples with RNA Integrity Number (RIN) > 5.5 (median 7.1; range 5.6–8.9) was performed as described in the TruSeq Stranded mRNA Library Prep Kit (Illumina, RS-122-2101). Briefly, 0.7–1 µg total RNA PolyA+ fraction was purified and randomly fragmented, converted to double stranded cDNA and processed through subsequent enzymatic treatments of end-repair, dA-tailing, and ligation to adapters. Adapter-ligated library was completed by PCR with Illumina PE primers. The resulting purified cDNA library was applied to an Illumina flow cell for cluster generation and sequenced on an Illumina HiSeq2500 on a 51 bp single-read format following manufacturer's protocols.

cDNA libraries from FFPE tumors and low integrity RNA's (RIN < 5.5) from FF tumors (median 2.4; range 1–5.5) but % of fragments > 200 nucleotides higher than 20% (median 57.5; range 20–96) were prepared using QuantSeq 3′ mRNA-Seq Library Prep Kit FWD for Illumina (Lexogen, 015) with a UMI Second Strand Synthesis Module for QuantSeq FWD (Lexogen, 081), following the vendor's protocol for low input/ low quality/ FFPE RNA. 200–500 ng of total RNA, according to availability, were used as starting material. PCR Add-on Kit for Illumina (Lexogen, 020) was used to adjust the library amplification number of cycles. Libraries were applied to an Illumina flow cell for cluster generation and sequenced on NovaSeq6000.

Image analysis, per-cycle basecalling and quality score assignment was performed with Illumina Real Time Analysis software. Demultiplexing of BCL files to FASTQ format was performed with the bcl2fastq Software (Illumina).

## RNA-Seq analyses

HTSeq-Counts from CNIO series were generated with Nextpresso[92] and BlueBee® Genomics Platform (Lexogen, 090-094), from FASTQ files obtained from TruSeq and QuantSeq platforms,

respectively. QuantSeq FWD-UMI Data Analysis Pipeline was used specifically for libraries constructed using UMIs. TopHat v2.1.1[93] and STAR v2.5.2a[94] algorithms were used to map the reads respectively to the reference genome GRCh37. RNA-Seq HTSeq-Counts for TCGA cohort were downloaded from UCSC Xena Browser (https://xenabrowser.net/).

HTSeq-counts from different platforms were integrated by common gene symbol collapsing by maximum expression value per gene within each platform. Normalized counts were used to infer tumor purity for each sample using the estimate score provided by ESTIMATE algorithm v1.0.13[36]. Samples failing quality control (e.g., low number of reads) or with ESTIMATE score greater than 5900 were excluded. Batch effects due to the type of analysis (UMI and no UMI), platform (QuantSeq and TruSeq), and cohorts (CNIO and TCGA) were corrected using ComBat[95] from R/Bioconductor package *sva*[96] v3.26.0 on log-transformed values. Samples considered outliers according to PCA, and from the consensus and hierarchical clustering were removed from further analysis.

### Genomic subtype classifier
The list of subtype-specific differentially expressed genes established by Burnichon et al.[20] was used to perform an unsupervised analysis of CNIO samples, and classify wild-type (WT) tumors into one of 3 genomic subtypes (pseudohypoxic, kinase signaling or Wnt-altered). Samples with known genotype were classified in one of 3 genomic subtypes as already defined[5].

### mPPGL signature discovery and validation
Differential expression analysis was executed between non-metastatic and primary metastatic tumors within each genomic subtype with DESeq2[97] v1.18.1 using ESTIMATE score as a covariate. Best candidates were ranked, and only those with |log2 fold-change| > 0.75 in all analyses and FDR < 0.01 in differential expression analysis including all cases were selected. A cohort of previously published PPGLs[20] was used for validation; only those cases with clinical behavior status available (extracted from[6]) were included, and updated genotype and genomic subtype classification were obtained from[98]. Data was analyzed as detailed elsewhere[29]. String v11 (https://string-db.org/) was used to perform a functional enrichment analysis of this gene signature.

### Gene set enrichment analysis
GSEA pre-ranked analysis was executed with GSEA software[99] v2.2.2 (RRID:SCR_003199), using either the selected gene sets obtained from the mutational/ SCNA functional enrichment analysis or the 'H: hallmark gene sets', 'C2: curated gene sets' and 'C5: ontology gene sets' collections from the Molecular Signature Database (MSigDB; https://www.gsea-msigdb.org/gsea/msigdb/). Default settings were used except for collapse set to false, scoring_scheme set to classic, plot_top_x set to 100, set_max set to 1000, and set_min set to 10. Input files were obtained from the differential expression analysis (non-metastatic vs primary metastatic tumors) and ranked according to the log2FoldChange.

### Classifier of metastasis prediction
The individual variables to be evaluated were dichotomized as done for the previous analyses and described in each figure legend. The total number of possible classifiers was dichotomized by computing all possible combinations with the 9 events to be evaluated ($n = 511$). We set as 1 - when any of the evaluated events included in the classifier was present; 0 - when none of the events was present. This analysis was executed using the dichotomized matrix with the samples for which we had all the data available ($n = 156$). We calculated the area under the receiver operating characteristic curves (AUC under ROC) of each event individually and all combinations possible ($n = 511$).

### Integrated analysis of focal SCNA and gene expression
A Fisher exact test was applied to establish differing focal SCNA, either gains or deletions, between non-metastatic and metastatic primary tumors at the gene level. Gains were based on DUP and DUP-LOH FACETS annotations, and deletions on HEMYZIG and DEL ones. 2619 genes appeared with a $P < 0.05$. Correlation between CN at gene level and expression data per gene was assessed using an anova test, selecting those with FDR < 0.05. After recoding FACETS output into four different values to facilitate analysis (DUP and DUP-LOH coded as 1, no event as 0, HEMYZIG as −1 and DEL as −2), Pearson's correlation coefficient between the CN scores and the expression levels was calculated. We identified significant genes by an Anova test and selected only those that exhibited a positive correlation (6785 genes). Only genes overlapping between both lists (911 genes) were considered further, assuming that these were genes with differing focal SCNA between studied groups and with a functional consequence at the mRNA level. Analyses were carried out in R (v.4.0.3).

### Mutational and SCNA functional enrichment analyses
Panther tool[25] was used to detect the significantly enriched gene sets among the consensus list of genes altered with SNV/INDELs or SCNAs. The SNV/INDEL gene list included all those genes with high impact variants (category 7: truncating, splice site, affecting start/stop codon variants, and missense variants classified as deleterious in three in silico predictors or in COSMIC, https://cancer.sanger.ac.uk/cosmic) in metastatic primary tumors. The SCNA gene list included those obtained through the *integrated analysis of focal SCNA and gene expression*.

Enriched annotations were retrieved from 'GO biological process', 'GO molecular function', 'GO cellular component' and 'Reactome pathways' using the Fisher's Exact test executed by Panther. For the consensus gene set enrichment analysis, a fold-enrichment (FC) (FDR < 0.05) threshold at 1.8 was established.

### Estimation of immune cell types
DESeq2 normalized expression matrix was used to calculate the abundance of 22 types of immune cell subsets in each sample of the CNIO cohort using CIBERSORTx[37], and the gene signature matrix LM22, 1000 permutations, B-mode batch correction, and disabled quantile normalization. Cell abundances from TCGA cohort were retrieved from Thorsson et al.[35]. Both matrices were transformed to z-scores (centered at the relative abundance mean of the primary metastatic samples of each cohort) and merged in a single matrix for further analysis.

### Immune subtype identification
We utilized the sample assignments established by Thorsson et al.[35] for the TCGA cohort. For CNIO cohort, we used the ImmuneSubtype-Classifier of the Cri-iAtlas[100] through the R package available in v0.1.0.

### Single-sample gene set enrichment analysis for TME subtype identification
Gene set variation analysis (GSVA) enrichment scores for the 29 curated gene signatures (Fges) from Bagaev et al.[38] were calculated using the GSVA R package[101] on the RNA-Seq expression matrix normalized with the VST function of DESeq2. Thereafter, we classified all tumors with RNA-Seq available into the 4 separate TME subtypes described by Bagaev et al. (Immune-enriched fibrotic, immune-enriched non-fibrotic, fibrotic, depleted) applying unsupervised k-means clustering (Euclidian distance, 1000 iterations) based on the 29 calculated Fges enrichment scores using Morpheus (https://software.broadinstitute.org/morpheus).

### Identification of neoantigens
Manually curated somatic SNVs and indels from the CNIO and TCGA cohort were annotated with VEP v106 using the required and useful

VEP options specified in the pVACtools[102] documentation. Coverage information for variant sites was retrieved from tumor and normal bam files using bam-readcount (Larson. genome/bam-readcount. GitHub https://github.com/genome/bam-readcount). For this purpose, the Docker 1.1.1 image of mgibio/bam_readcount_helper-cwl was used and the collected data were added to the annotated VFC files with the vcf-readcount-annotator from the VAtools package (http://vatools.org). HLA class I typing was performed on each sample with POLYSOLVER[103]v4. pVACseq from the pVACtools package version 3.0.2 was then used for epitope prediction. The epitope length was set to 8–11 and the prediction algorithms were all those available for MHC Class I (NetMHCpan, NetMHC, NetMHCcons, PickPocket, SMM, SMMPMBEC, MHCflurry and MHCnuggetsI). The epitopes provided by pVACseq in the filtered files, which are those passing binding affinity and sequence-based thresholds were the ones used to establish the neoantigen load per sample.

### Immunohistochemistry

Antigen retrieval was first performed on 3 μm FFPE sections with the appropriate pH buffer and endogenous peroxidase was blocked (peroxide hydrogen at 3%). Slides were incubated with the appropriate primary antibody as detailed: mouse anti-CDK1 Ab (dil 1:250; BD Biosciences Cat# 610038, RRID:AB_397454), rabbit anti-PDL1 [E1L3N] XP mAb (dil 1:150 20'ER2; Cell Signaling Technology Cat# 13684, RRID:AB_2687655) and rat anti-CD8A [NOR132H] (dil 1:20; CNIO in-house Ab validated by EuroMAbNet; https://www.euromabnet.com/monoclonal-antibodies/cd8a/38.html). Afterwards, the corresponding visualization systems was used when needed (Bond Polymer Refine Detection, Bond, Leica; EnVision FLEX+, Dako) conjugated with horseradish peroxidase. Immunohistochemical reaction was developed using 3,30-diaminobenzidine tetrahydrochloride (DAB) (Dako). Nuclei were counterstained with Harrys's hematoxylin. Finally, the slides were dehydrated, cleared and mounted with a permanent mounting medium for microscopic evaluation. Whole slides images were scanned with AxioScan Z1 (Zeiss), and captured either with a Zen Blue software (ZEN Digital Imaging for Light Microscopy, Zeiss, RRID:SCR_013672) or AxioVision software (AxioVision Imaging System, RRID:SCR_002677).

For CDK1 staining quantification, a script was created using Zen Blue software (additional module for analysis, Zeiss); positivity was stablished as stained area in tumor area after manual tumor selection on hematoxylin and eosin-stained slides. After training and script optimization, the quantification program was run and results exported as excel files with scoring data for each image file.

PD-L1 staining was assessed by two independent observers (G. Roncador and B. Calsina) on complete sections. Each case was scored semi-quantitatively depending on the tumor proportion score (TPS), as tier 1 (0–15% positive tumor cells), tier 2 (15–30%), and tier 3 (30–100%).

For CD8+ T cells quantification, a digital image analysis workflow was developed in QuPath[104]. Briefly, batch analysis was applied across all whole slides to automatically perform cell detection and count the number of positive cells. The cell detection parameters were optimized by applying the following parameters: optical density sum detection image, pixel size: 0.5 μm, background radius: 18 μm, median radius: 2 μm, minimum/maximum area: 12–1000 μm, intensity threshold: 0.07, sigma: 1.3 μm, threshold compartment: *Cell:DAB OD mean* 0.26. A random tree (RTees) classifier was trained for each tumor slide in order to distinguish between tumor cells and normal adrenal, sustentacular and endothelial cells, and erythrocytes. Manual pathologist (E. Caleiras) assessment was used to validate the classifier in a subset of cases.

### Quantification and statistical analysis

Statistical analyses were performed either using R version 3.2.2 (RRID:SCR_000432) or IBM SPSS Statistics v19 (RRID:SCR_019096). The statistical details of experiments including the exact value of $n$ in terms of number of samples for a given cohort, the experimental method, controlling/covariates used and specific statistical tests employed are reported in the relevant section (Results, Figures and Figure Legends, Supplementary Tables). For a given test, significance was defined, unless expressly indicated, if a $p$ value was less than 0.05. Disease-specific progression-free was established as the time interval from the PPGL primary diagnosis to the first appearance of a metastatic lesion (time to progression, TTP), as a reliable prognostic measure.

### Reporting summary

Further information on research design is available in the Nature Portfolio Reporting Summary linked to this article.

## Data availability

The WES and RNA-Seq fastq files generated during this study, as well as level 3 files (expression count matrix, and the VCF files with the variant calling results for both SNV/INDELs and CN) have been deposited in the European Genome-Phenome Archive (EGA) under the accession EGAS00001006043 and EGAS00001006044. The data are available under restricted access due to the possibility of revealing patient-sensitive information. Request for data access will be referred directly to the Data Access Committee (DAC) of the CNIO (mrobledo@cnio.es). The access will be granted for health/medical/biomedical purposes and according to good practice recommendations. The DAC will attempt to provide a response to all applications within two weeks of submission and render a final decision within no more than four weeks. Once access has been granted, data will be available for one month. The remaining data are available within the article and supplementary information. The publicly available microarray dataset used in this study is available from ArrayExpress (E-MTAB-733)[20] and the TCGA WES and RNA-Seq data from GDC Data Portal, NIH (https://portal.gdc.cancer.gov/), TCGA-PCPG[5]. Source data are provided with this paper.

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

## Acknowledgements

This work was supported by Project PI17/01796 and PI20/01169 to M.R. [Instituto de Salud Carlos III (ISCIII), Acción Estratégica en Salud, cofinanciado a través del Fondo Europeo de Desarrollo Regional (FEDER)], Paradifference Foundation [no grant number applicable to M.R.], Pheipas Association [no grant number applicable to M.R.], the Clinical Research Priority Program of the University of Zurich for the CRPP HYRENE to F.B., the Deutsche Forschungsgemeinschaft (DFG) within the CRC/Transregio 205/1 (Project No. 314061271-TRR205 to to F.B., M.F., N.B., and G.E.) and the Instituto de Salud Carlos III (ISCIII), Spanish Ministry of Science and Innovation (Project No. PID2019-111356RA-I00 to G.M.). B.C. was supported by the Rafael del Pino Foundation (Becas de Excelencia Rafael del Pino 2017). A.M.M.-M. was supported by CAM (S2017/BMD-3724; TIRONET2-CM). A.F.-S. and J.L. received the support of a fellowship from La Caixa Foundation (ID 100010434; LCF/BQ/DR21/11880009 and LCF/BQ/DR19/11740015, respectively). M.M., S.M., and M.S. were supported by the Spanish Ministry of Science, Innovation and Universities "Formación del Profesorado Universitario— FPU" fellowship with ID number FPU18/00064, FPU19/04940 and FPU16/05527. A.D.-T. is supported by the Centro de Investigacion Biomédica en Red de Enfermedades Raras (CIBERER). L.J.L.-G. was supported both by the Banco Santander Foundation and La Caixa Postdoctoral Junior Leader Fellowship (LCF/BQ/PI20/11760011). C.M.-C. was supported by a grant from the AECC Foundation (AIO15152858 MONT). We thank the Spanish National Tumor Bank Network (RD09/0076/00047) for the support in obtaining tumorsamples and all patients, physicians and tumor biobanks involved in the study.

## Author contributions

Conceptualization, B.C., E.P.Y., A.M.M.M., F.A., and M.R.; Methodology, B.C., E.PY., A.M.M.M., E.C., E.G., O.D., G.R., J.F.G.G., F.A., and M.R.; Software, E.P.Y., A.M.M.M., R.T.P., and C.F.T.; Formal analysis, B.C. and S.G.M.; Investigation, B.C., E.C., A.F.S., M.M., M.P.A., R.L., S.J., M.C.M., J.M.R.R., J.L., M.S., A.D.T., A.R., P.G., S.R.P., and C.M.C.; Resources, N.B., M.D., M.C., S.G., C. A.E., R.M.R., J.A., M.I.O.G., A.L.F., S.M.J.F., E.R., M.F., F.B., M.Q., M.M., H.J.T., G.E., and K.P.; Data curation, B.C., M.C.F., and A.C.; Writing – original draft, B.C., E.P.Y., and A.M.M.M.; Writing – review & editing, B.C., E.P.Y., A.M.M.M., A.F.S., M.M., J.L., S.M., B.H., M.F., F.B., R.A.T., G.E., O.D., G.M., C.R.A., C.M.C., A.C., L.J.L.G., K.P., F.A., and M.R.; Visualization, B.C.; Supervision, B.C., F.A., and M.R.; Funding acquisition, M.R.

## Competing interests

G.M., is a founder, director and shareholder of Tailor Bio Ltd, a genomics company using copy number signatures for precision medicine. The rest of the authors declare no competing interests.

## Additional information

**Bruna Calsina** ⑩ [1] ✉, **Elena Piñeiro-Yáñez** ⑩ [2,31], **Ángel M. Martínez-Montes** ⑩ [1,31], **Eduardo Caleiras** [3], **Ángel Fernández-Sanromán** ⑩ [1], **María Monteagudo** ⑩ [1], **Rafael Torres-Pérez** ⑩ [1,4], **Coral Fustero-Torre** ⑩ [2], **Marta Pulgarín-Alfaro** [1], **Eduardo Gil** [1], **Rocío Letón** [1], **Scherezade Jiménez** [5], **Santiago García-Martín** ⑩ [2], **Maria Carmen Martin** [6], **Juan María Roldán-Romero** [1], **Javier Lanillos** ⑩ [1], **Sara Mellid** [1], **María Santos** [1], **Alberto Díaz-Talavera** ⑩ [1,7], **Ángeles Rubio** [8], **Patricia González** [3], **Barbara Hernando** [9], **Nicole Bechmann** ⑩ [10], **Margo Dona** [11], **María Calatayud** [12], **Sonsoles Guadalix** [12], **Cristina Álvarez-Escolá** [13], **Rita M. Regojo** [14], **Javier Aller** [15],

Maria Isabel Del Olmo-Garcia[16], Adrià López-Fernández [17], Stephanie M. J. Fliedner [18], Elena Rapizzi[19], Martin Fassnacht [20,21], Felix Beuschlein [22,23], Marcus Quinkler[24], Rodrigo A. Toledo [25,26], Massimo Mannelli[19], Henri J. Timmers[11], Graeme Eisenhofer[10,27], Sandra Rodríguez-Perales [6], Orlando Domínguez[8], Geoffrey Macintyre [9], Maria Currás-Freixes[1,28], Cristina Rodríguez-Antona[1,7], Alberto Cascón [1,7], Luis J. Leandro-García[1], Cristina Montero-Conde [1,7], Giovanna Roncador[5], Juan Fernando García-García[29], Karel Pacak [30], Fátima Al-Shahrour [2] & Mercedes Robledo [1,7] ✉

[1]Hereditary Endocrine Cancer Group, Human Cancer Genetics Program, Spanish National Cancer Research Centre (CNIO), Madrid, Spain. [2]Bioinformatics Unit, Structural Biology Program, Spanish National Cancer Research Centre (CNIO), Madrid, Spain. [3]Histopathology Core Unit, Biotechnology Program, Spanish National Cancer Research Centre (CNIO), Madrid, Spain. [4]Bioinformatics for Genomics and Proteomics, National Centre for Biotechnology (CNB-CSIC), Madrid, Spain. [5]Monoclonal Antibodies Core Unit, Biotechnology Program, Spanish National Cancer Research Centre (CNIO), Madrid, Spain. [6]Molecular Cytogenetics and Genome Engineering Group, Human Cancer Genetics Program, Spanish National Cancer Research Centre (CNIO), Madrid, Spain. [7]Centro de Investigación Biomédica en Red de Enfermedades Raras (CIBERER), Institute of Health Carlos III (ISCIII), Madrid, Spain. [8]Genomics Core Unit, Biotechnology Program, Spanish National Cancer Research Centre (CNIO), Madrid, Spain. [9]Computational Oncology Group, Structural Biology Program, Spanish National Cancer Research Centre (CNIO), Madrid, Spain. [10]Institute of Clinical Chemistry and Laboratory Medicine, University Hospital Carl Gustav Carus, Technische Universität Dresden, Dresden, Germany. [11]Department of Internal Medicine, Radboud University Medical Centre, Nijmegen, The Netherlands. [12]Department of Endocrinology, 12 de Octubre University Hospital, Madrid, Spain. [13]Department of Endocrinology, La Paz University Hospital, Madrid, Spain. [14]Department of Pathology, La Paz University Hospital, Madrid, Spain. [15]Department of Endocrinology, Puerta de Hierro University Hospital, Madrid, Spain. [16]Department of Endocrinology and Nutrition, University Hospital La Fe, Valencia, Spain. [17]Department of Medical Oncology, Vall d'Hebrón Hospital, Barcelona, Spain. [18]Neuroendocrine Oncology and Metabolism, Medical Department I, Center of Brain, Behavior, and Metabolism, University Medical Center Schleswig-Holstein Lübeck, Lübeck, Germany. [19]Department of Experimental and Clinical Biomedical Sciences, University of Florence, Florence, Italy. [20]Department of Internal Medicine I, Division of Endocrinology and Diabetes, University Hospital Würzburg, University of Würzburg, Würzburg, Germany. [21]Comprehensive Cancer Center Mainfranken, University of Würzburg, Würzburg, Germany. [22]Medizinische Klinik und Poliklinik IV, Klinikum der Universität München, Munich, Germany. [23]Klinik für Endokrinologie Diabetologie und Klinische Ernährung, Universitätsspital Zürich (USZ) und Universität Zürich (UZH), Zürich, Switzerland. [24]Endocrinology in Charlottenburg Stuttgarter Platz 1, Berlin, Germany. [25]Gastrointestinal and Endocrine Tumors, Vall d'Hebron Institute of Oncology (VHIO), Vall d'Hebron Barcelona Hospital Campus, Barcelona, Spain. [26]Centro de Investigación Biomédica en Red de Cáncer (CIBERONC), Institute of Health Carlos III (ISCIII), Madrid, Spain. [27]Department of Medicine III, University Hospital Carl Gustav Carus, Technische Universität Dresden, Dresden, Germany. [28]Department of Endocrinology, Clínica Universidad de Navarra, Madrid, Spain. [29]Department of Pathology, MD Anderson Cancer Center, Madrid, Spain. [30]Section of Medical Neuroendocrinology, Eunice Kennedy Shriver National Institute of Child Health and Human Development, National Institutes of Health, Bethesda, MD, USA. [31]These authors contributed equally: Elena Piñeiro-Yáñez, Ángel M Martínez-Montes. ✉e-mail: bcalsina@cnio.es; mrobledo@cnio.es

