## [Peer Review File · Nature Communications]

GENOMIC AND IMMUNE LANDSCAPE OF METASTATIC PHEOCHROMOCYTOMA AND PARAGANGLIOMAReviewers' Comments:

Reviewer #1:

Remarks to the Author:

Dr Calsina and co-authors have performed whole-exome sequencing of 87 and RNA sequencing of 114 PPGLs, of which a staggering 99 samples were metastatic (at initial presentation or on follow-up). The mutational landscape is presented, with a heavy focus on MSI and tumor mutational burden (TMB). Comparisons to the TCGA database have been performed as well. Higher MSI and TMB were associated to metastatic events, Wnt altered PPGLs, tumor size and Ki-67 indices >5%. Moreover, ATRX mutations and TERT gene aberrancies (gene expression, promoter mutations, promoter hypermethylation and/or copy number gain) were also associated to metastatic events and higher TMB/MSI, mainly in Wnt altered and pseudohypoxic driven tumors. Metastatic PPGLs also carried the highest number of copy number alterations. Interestingly, gain of ch. 5p (including the TERT gene locus) was independently associated to primary tumors that subsequently metastasized. A gene expression signature was furthermore associated to metastatic disease, and CDK1 stood out as the most significantly altered gene in terms of expression. As this observation was reproduced using immunohistochemistry, the authors pinpoint the potential use of this predictive marker in clinical routine. In addition to this, the tumoral immune landscape of the PPGL cohort was detailed using the RNA seq data to generate immune profiles - and metastatic primary tumors and metastases were more often classified as C4 ("lymphocyte depleted") than non-metastatic cases, and had lower levels of CD8+ T-killer cells. PD-L1 immunexpression was especially pronounced in Wnt altered tumors, whereas the levels were lowest among pseudohypoxic PPGLs.

The manuscript is well-written, and illustrations are lucid and motivated. The WES and RNA seq pipelines and filtering processes are adequately described. While the work is only partly original, as MSI and TMB have been previously characterized in PPGL, the comprehensive characterization of the immune landscape and the sheer size of the cohort make the work clinically motivated. The coupling between genotype clusters, TMB/MSI, high-risk alterations (ATRX/TERT) also make clinical sense, and therefore renders the association to different immune response profiles significant from an oncology perspective. As of this, I believe the data presented in here represents a considerable contribution to the field. I do however have several queries for the authors to consider:

Major points:

1. Given the implementation of the 2022 WHO classification of endocrine and neuroendocrine tumors, some rare PPGL subtypes are no longer classified as paraganglioma, including gangliocytic PGLs and cauda equina PGLs. I could not find information regarding the anatomic localization of each PGL in this cohort, so please make sure this submission is updated accordingly. In line with this, reference #73 is obsolete and should be replaced by PMID: 35285002.
2. The authors make use of the terminology "aggressive", and define PPGLs as aggressive "when there was capsular or adipose tissue invasion, vascular infiltration or evidences of multiple recurrences, without certainty of metastatic disease". While I agree on some of these parameters (supported by PASS/GAPP histological triaging algorithms etc.), isolated capsular invasion is recurrently seen in large PPGL series with associated molecular data (PMID: 30451732), and very seldomly coupled to subsequent metastatic behavior. While I appreciate that the authors are trying to single out cases with histological worrisome features, I still think the parameters used for "aggressive" should be motivated based on previous studies. Also, sometimes these cases are referred to as "clinically aggressive" in figures (Figure 2 etc.), please streamline.
3. Please specify how the tissue from the metastatic PPGLs were collected. Surgical excision? Core needle biopsies? I just want to make sure the results are not biased by decalcification reagents (21% of cases were bone metastases), and also please state if any of these patients have been treated in

any kind prior to the extraction of the metastatic tissue (chemo/radiotherapy/targeted therapy). If yes, could this have affected the MSI/TMB results?

4. The authors suggest the rapid implementation of CDK1 IHC in clinical routine in order to pinpoint metastatic potential of PPGLs. However, the authors made use of an automated scoring algorithm which I doubt would be universally accepted without extensive validation. The scoring should therefore be presented using standardized eye-balling methodology, also reporting the subcellular localization. Please also provide the sensitivity and specificity for this marker in ruling in future metastatic events. I assume already metastatic cases were excluded from this IHC?

5. In this manuscript, the PD-L1 immunoreactivity is reported in percentage of tumor cells, and was semi-quantitatively scored in three tiers. While not much is known regarding PD-1 blockade in PPGLs, I still wonder why the authors did not report the combined positive score (CPS) used for other solid tumors? Please explain why PD-L1 expression in tumor-infiltrating lymphocytes was disregarded in this aspect. Also, the range for "negative" is 0-15% of tumor cells (row 764). This range is too wide, most studies report <1% or above 1% and beyond - in which response to PD-1 blockade can be achieved in tumors with for example CPS >5%.

6. Could the immunophenotyping results (ImmunoScore etc.) obtained from the metastases differ depending on metastatic site? And if so, could the different metastatic sites affect the interpretation of the analyses? One would assume that the immune response should differ between a lymph node metastasis and a bone metastasis?

7. Just by eyeballing Figure 1, an IHC panel consisting of ATRX, Ki-67 (5% cut-off) and SDHB would potentially be adequate in identifying many of the PPGLs with high TMB/MSI, and probably have high sensitivity for most metastatic cases. Since these markers are routinely available in most pathology laboratories (while CDK1 is not), please comment on the potential of the former. I assume there are no established clinical routine analyses for interrogating TERT expression?

Minor points:

8. What is "normal tissue" in the context of PPGL, do the authors refer to normal adrenal tissue or normal paraganglia? This is crucial information. If normal adrenal medullae were assessed, please comment on the discrepancies compared to paraganglia.

9. Discussion: "...the molecular switches involved in metastatic progression are likely variable and involve non-genetic events." Since the authors are not interrogating the cohort via WGS, but rather WES+RNA seq, there is still a possibility that the events driving metastatic spread dwell outside of the exome.

10. Row 624: "hypermethylation" is misspelled.

11. Figure 6C: "no-metastatic"  "non-metastatic".

Reviewer #2:

Remarks to the Author:

Calsina et al. describe the results obtained by the genomic profiling of a large cohort of mPPGLs. They integrate their data with those of the TCGA to compare metastatic vs non-metastatic PPGL.

Comparison of MSI score and TBM show some differences between PPGL and mPPGL. Despite the various detailed analyses performed by the group, there is no clear genomic hallmark that predicts metastatic vs non-metastatic diseases. However, the study is of interest to the community since the data generated can be used for secondary studies. Moreover, the reported immune characterization is

the main interesting part but could be further expanded.

Major points:

1. The authors mention that they have a good number of tumors ($16+4=20$) for which there is more than one tumor sequenced by WES. However, this is not exploited in the paper. Are these multiregional samples, just replicates or longitudinal samples? Even if the number is small, if they have longitudinal samples the author should try to exploit these data. Even if these cases are multiregional, the evaluation of heterogeneity and subclonal analysis could help to better elucidate the heterogeneity of this disease. The authors could try to better describe and exploit these tumors.
2. The transcriptomic analysis is based on the integration of different platforms and different dataset using combat. If I understand correctly, the supervised analysis of Fig 4a is performed on the integrated dataset. Could the relatively low number of differentially expressed (DE) genes depend on the over-smoothing induced by Combat? Did the authors try to separately perform supervised analysis between mPPGLs and PPGLs? It seems from Figure ED1b that it would be possible. The validation of ED5a reports just the 27 genes of Figure 4a. What about all the DE genes in this validation dataset? Overall, it seems that the transcriptomic signature is not robust enough and the authors should try to better validate the signature. However, the prognostic role of CDK1 is convincing, but not completely novel.
3. The immune landscape is interesting, but it is not related to the metastatic PPGL, rather it confirms the heterogeneity of mPPGLs and the lack of differences. Also, the TME subtypes are loosely associated with the genomics subtypes with the exception of pseudohypoxic in the Fibrotic TME subtype. Figure 7c could be done differentiating metastatic versus non-metastatic.
4. The data of figure 7e showing the greater infiltration of CD8+ T cells in MAML3-tumors can be interesting but is limited few cases. Why this was not detected using the Cybersort deconvolution?
5. Related to the previous point, the authors describe the TME classification of PPGLs using the large dataset and the integrated transcriptomic. However, there is no downstream analysis of these clusters. Are there any genomics events associated with these TME clusters? Is there any enrichment of genomics drivers in one cluster vs the other? What about the MSI score and TBM that you have reported in terms of the genomics clusters?
6. If the authors want to describe an immune landscape of mPPGLs, that part of the paper completely lacks the study of the neoantigens. I suggest, in addition to the TMB, characterizing the neoantigen load in the four TME subtypes, and interpreting immuno-editing events.
7. Fastq files have been deposited on EGA (but still not available). If this study wants to complements the TCGA for mPPGLs, the authors should definitely share level 3 data (count matrices, MAF files, copy number segment files).

REVIEWER COMMENTS

Reviewer #1, expertise in PPGL subtypes/WES/RNA-seq (Remarks to the Author):

Dr Calsina and co-authors have performed whole-exome sequencing of 87 and RNA sequencing of 114 PPGLs, of which a staggering 99 samples were metastatic (at initial presentation or on follow-up). The mutational landscape is presented, with a heavy focus on MSI and tumor mutational burden (TMB). Comparisons to the TCGA database have been performed as well. Higher MSI and TMB were associated to metastatic events, Wnt altered PPGLs, tumor size and Ki-67 indices >5%. Moreover, ATRX mutations and TERT gene aberrancies (gene expression, promoter mutations, promoter hypermethylation and/or copy number gain) were also associated to metastatic events and higher TMB/MSI, mainly in Wnt altered and pseudohypoxic driven tumors. Metastatic PPGLs also carried the highest number of copy number alterations. Interestingly, gain of ch. 5p (including the TERT gene locus) was independently associated to primary tumors that subsequently metastasized. A gene expression signature was furthermore associated to metastatic disease, and CDK1 stood out as the most significantly altered gene in terms of expression. As this observation was reproduced using immunohistochemistry, the authors pinpoint the potential use of this predictive marker in clinical routine. In addition to this, the tumoral immune landscape of the PPGL cohort was detailed using the RNA seq data to generate immune profiles - and metastatic primary tumors and metastases were more often classified as C4 ("lymphocyte depleted") than non-metastatic cases, and had lower levels of CD8+ T-killer cells. PD-L1 immunexpression was especially pronounced in Wnt altered tumors, whereas the levels were lowest among pseudohypoxic PPGLs.

The manuscript is well-written, and illustrations are lucid and motivated. The WES and RNA seq pipelines and filtering processes are adequately described. While the work is only partly original, as MSI and TMB have been previously characterized in PPGL, the comprehensive characterization of the immune landscape and the sheer size of the cohort make the work clinically motivated. The coupling between genotype clusters, TMB/MSI, high-risk alterations (ATRX/TERT) also make clinical sense, and therefore renders the association to different immune response profiles significant from an oncology perspective. As of this, I believe the data presented in here represents a considerable contribution to the field. I do however have several queries for the authors to consider:

Major points:

1. Given the implementation of the 2022 WHO classification of endocrine and neuroendocrine tumors, some rare PPGL subtypes are no longer classified as paraganglioma, including gangliocytic PGLs and cauda equina PGLs. I could not find information regarding the anatomic localization of each PGL in this cohort, so please make sure this submission is updated accordingly. In line with this, reference #73 is obsolete and should be replaced by PMID: 35285002.

R/ We thank the Reviewer for this observation. We have reviewed the localization of the PGLs included, and now we can state that non gangliocytic or cauda equina PGL were included in the series. We have now incorporated the data regarding anatomic localization in Supplementary Table 1. We have also updated the reference as proposed (line 511).

2. The authors make use of the terminology "aggressive", and define PPGLs as aggressive "when there was capsular or adipose tissue invasion, vascular infiltration or evidences of multiple recurrences, without certainty of metastatic disease". While I agree on some of these parameters (supported by PASS/GAPP histological triaging algorithms etc.), isolated capsular invasion is recurrently seen in large PPGL series with associated molecular data (PMID: 30451732), and very seldomly coupled to subsequent metastatic behavior. While I appreciate that the authors are trying to single out cases with histological worrisome features, I still think the parameters used for "aggressive" should be motivated based on previous studies. Also, sometimes these cases are referred to as "clinically aggressive" in figures (Figure 2 etc.), please streamline.

R/ From the 10 tumors classified as aggressive (all of them profiled by RNA-Seq, and 3 also by WES), none was classified in this group only due to isolated capsular invasion. Please, see below the reasons why each of the tumors were classified as aggressive:

- for three, the patients had multiple recurrences
- for three, vascular invasion was described
- for two, capsular, vascular and adipose tissue invasion was reported
- for one, capsular and vascular invasion was noted
- and for another one, the patient had multiple recurrences, and also adipose tissue invasion was described

"Clinically aggressive" has now just been corrected as "aggressive" in the text and figures accordingly to Reviewer's comment. Also, we now describe aggressive tumors in the method section (line 511-513) as: "Aggressive disease was established when there was capsular or adipose tissue invasion, and vascular infiltration or evidences of multiple recurrences, without certainty of metastatic disease".

3. Please specify how the tissue from the metastatic PPGLs were collected. Surgical excision? Core needle biopsies? I just want to make sure the results are not biased by decalcification reagents (21% of cases were bone metastases), and also please state if any of these patients have been treated in any kind prior to the extraction of the metastatic tissue (chemo/radiotherapy/targeted therapy). If yes, could this have affected the MSI/TMB results?

R/ In the clinical records of the CNIO series, it is noted that 21 metastatic tissues were collected by surgical excision (1 of them was a bone metastasis), 3 of them by core needle biopsy (all of them bone metastases) and from 5 tissues we did not have information of the collection method. The Anatomic Pathology Departments of the hospitals of origin of the metastases were contacted and we have confirmed the methods used to decalcify take into account this potential problem. It was used either soft decalcifier EDTA bisodium in acid

buffer, or decalcifiers similar to those used in bone marrow decalcification. The decalcification was slow and accompanied by tissue fixation.

Regarding the treatment received before sample collection, we have the information of 61% of the patients from whom metastases were collected. From those, only 3 patients received radiotherapy before the collection of the tissues used for this study. None of the metastases included in the study came from patients receiving chemotherapy or targeted therapies before the collection of their tissues. None information is available for the 2 metastases included in the TCGA study. Fig R1 shows TMB and MSI scores in patients untreated and treated with radiotherapy before tumor sample collection. TMB do not seem to be affected, but we do observe a tendency of lower MSI scores in tumors coming from patients treated with radiotherapy. One could expect the contrary, as radiation therapy directly affects DNA structure by inducing DNA breaks, and therefore potentially increasing MSI score lectures. Therefore, and also taking into account that we only have 3 metastases radiotherapy treated, we believe that this variable is not biasing nor TMB neither MSI score results. This is the reason why we are maintaining these tumors in the study. Moreover, as these are the ones presenting the lowest MSI scores, if they were excluded from the study, the differences of the metastases in comparison to the other tumor types would be even more accentuated in Fig 2c. If the Reviewer still considers that these 3 samples should be excluded, we would be willing to do so.

Fig R1. (a) TMB and (b) MSI score (extracted with MANTIS) across PPGL metastases. To test differences between not treated and radiotherapy-treated metastases a Mann-Whitney-Wilcoxon (MWW) two-sided test was used.

4. The authors suggest the rapid implementation of CDK1 IHC in clinical routine in order to pinpoint metastatic potential of PPGLs. However, the authors made use of an automated scoring algorithm which I doubt would be universally accepted without extensive validation. The scoring should therefore be presented using standardized eye-balling methodology, also reporting the subcellular localization. Please also provide the sensitivity and specificity for this marker in ruling in future metastatic events. I assume already metastatic cases were excluded from this IHC?

R/ We have tried to establish an eye-balling methodology to score the CDK1 IHC as the Reviewer proposed, and we can conclude that, at least in this small series, we are not able

to identify in a qualitative manner differences in the CDK1 levels between non-mPPGLs and m-PPGLs. Therefore, the Reviewer is right and we cannot state that CDK1 IHC could experiment a rapid implementation in clinical routine at the moment. To evaluate CDK1 IHC, a digital pathology method is needed to quantitatively measure the protein levels of CDK1. These methods are expected to enhance precision, reproducibility and be scalable by AI-powered analysis tools. However, what it is true is that there are still many challenges this technology may face if adopted in clinical settings (reviewed in Baxi, V. *Mod Pathol.* 2022). Consequently, we have eliminated the part of the text where we discussed about its rapid implementation.

We are certain that *CDK1* mRNA is overexpressed in mPPGLs, therefore we believe that *CDK1* mRNA expression quantification would be a quicker tool to be implemented in the clinical routine. The classification power of CDK1 expression has been evaluated by computing a receiver operating characteristic (ROC) curve. Continuous expression of *CDK1* in our cohort and in the independent PPGL cohort we used to validate some of our findings showed a high accuracy for this classifier (Fig R2). The sensitivity and specificity reported for dichotomized high CDK1 protein levels reported in Reviewer#1, question#7 using the series of tumors with all variables analyzed available was 64.7% and 85.2%, respectively

Fig R2. ROC curve analysis showing the accuracy of *CDK1* expression to discriminate mPPGL patients. All samples used in Fig 4d,e were included in the analysis.

The subcellular localization was nuclear and it has been added in the manuscript text (line 244-245) as follows: “We also validated by immunohistochemistry (IHC) higher CDK1 protein levels in the nucleus of primary metastatic tumors (Fig. 4f)”.

Regarding the Reviewer’s comment “I assume already metastatic cases were excluded from this IHC?”, we can state that metastases were not included in the analysis; only primary tumors that were classified as metastatic at diagnosis or during clinical follow-up. It is true that primary tumors from metastatic patients at diagnosis should not be included when defining a marker of risk of metastasis, but our IHC series was composed by only n=7 primary tumors from patients with metachronous metastatic disease, and therefore, currently we do not have enough statistical power for that type of analysis. However, a ROC curve analysis could be performed using the expression matrix which includes a sufficient number of metachronous cases. If synchronous cases are excluded from the analysis, the elevated

CDK1 expression for identifying metastatic cases shows even higher accuracy (AUC=0.807; 95% CI=0.712-0.903) than when synchronous cases are included (Fig R2), leading us to believe that if we had more metachronous cases in the IHC series we might see the same.

5. In this manuscript, the PD-L1 immunoreactivity is reported in percentage of tumor cells, and was semi-quantitatively scored in three tiers. While not much is known regarding PD-1 blockade in PPGLs, I still wonder why the authors did not report the combined positive score (CPS) used for other solid tumors? Please explain why PD-L1 expression in tumor-infiltrating lymphocytes was disregarded in this aspect. Also, the range for "negative" is 0-15% of tumor cells (row 764). This range is too wide, most studies report <1% or above 1% and beyond - in which response to PD-1 blockade can be achieved in tumors with for example CPS >5%.

Following the Reviewer's comment we have re-assessed using a semi-automatic digital image analysis workflow in QuPath the CPS taking into account whole slides. The Reviewer is right regarding the established range set for the "negative" tier, and therefore we have downgraded the "negative" tier to CPS<5%, set an "intermediate" (+) positivity (CPS=5-20%) and a "high" positivity to CPS>20% (Fig R3).

Fig R3. Quantification of PD-L1 IHC within **(a)** genomic subtype in a subset of n=44 PPGLs and **(b)** tumor behavior in a subset of n=41 PPGLs (three PPGLs classified as aggressive were omitted from the analysis). Two-sided Freeman-Halton test was used to test for differences between groups. P values shown in (a) are the ones obtained from the analysis of MAML3 class and the other groups. The other groups among them do not exhibit statistically significant differences.

Methods for assessing PD-L1 expression are known to vary slightly between assays, as do positive cut-off values for predicting response to immunotherapy, which can range from 1% to 100% depending on the test and cancer type. For patients who may benefit from PD-L1 testing, selecting the best test to evaluate the biomarker is not always straightforward. The FDA has validated and approved different immunohistochemical assays for different tumor types using different PD-L1 antibodies, cut-off values for positivity and scoring systems, including both Tumor Proportion Score (TPS) and CPS (reviewed in Li *et al. Br J Cancer.* 2022). The specific test needed to assess PD-L1 expression depends on the type of cancer and the specific immune checkpoint inhibitor therapy being considered. For example, in the single clinical trial in which PD-1 checkpoint inhibitor has been tested in mPPGLs (Jimenez C *et al. Cancers* 2020), Merck 22C3 antibody was used for PD-L1 staining. To evaluate these IHCs, an H-score ≥ 42.5 was used as cut-off value, even though not identifying an association between PD-L1 positivity and clinical response. Exploratory studies in PPGLs (Celada *et al. J Pathol.* 2022 and Guo *et al. Hum Pathol.* 2018) using the same PD-L1 antibody used in our study, established different scoring methods too. The first one states:

“PD-L1 immunostaining was evaluated as positive when all tumor cells or any percentage of them showed membrane immunostaining”, while the second one estimated positive cells on a scale of 0 to 100% by establishing a semiquantitative immunohistoscoring (IHS) as follows: “The average intensity of staining cells was given a score from 0 to 3 (0 = none; 1 = weak; 2 = intermediate; 3 = strong). The IHS was calculated by multiplying the percentage and intensity scores, and the continuous HIS, ranging from 0–300, was categorized around the median of the distribution. Cases with a focal pattern of PD-L1 expression of moderate intensity in more than 5% of the tumor cells were considered positive (HIS > 10)”.

All of this indicates that there is a clear need to establish a standardized method to assess PD-L1 IHC specifically for PPGLs, which should determine the antibody to be used, the cut-off value for positivity and the scoring system. Ultimately, to associate this standardized method to the clinical response to PD-1/PD-L1 checkpoint inhibitors, a protocol for validation of the method in a clinical trial should be included. For all these reasons, we believe that the scoring system chosen in this study is still acceptable, pending the establishment of a standardized method for PPGLs. However, we agree with the Reviewer on the fact that 0-15% cannot be called “negative” and, therefore, we have changed the nomenclature by indicating in Fig 7D, ED Fig 12b and in the methods section (line 809-810) the TPS range per tier. If the Reviewer still considers it necessary, we could replace Fig 7D and ED Fig 12b for Fig R3a,b.

6. Could the immunophenotyping results (ImmunoScore etc.) obtained from the metastases differ depending on metastatic site? And if so, could the different metastatic sites affect the interpretation of the analyses? One would assume that the immune response should differ between a lymph node metastasis and a bone metastasis?

R/ We agree with the Reviewer that the TME can differ depending on metastatic site, as it has been reported for other tumor types. We have now evaluated immunoscore, immune cell levels inferred with CIBERTSORTx and the expression of the presented immunomodulators (IM) from Fig 7c in the metastases included in the CNIO cohort with RNA-Seq data available. As expected, we observed a tendency of higher immunoscore in lymph nodes metastases (Fig R4a, MWW to test for differences between lymph nodes and other metastatic sites: $P=0.306$). Unsupervised hierarchical clustering based on the Euclidean distance matrix of Z scores of immune cell levels showed that metastases partially segregated by target organ, and revealed that metastases from the same patient (A and B) did show more similarities (Fig R4b). We also examined the expression of IM discovering that metastases continued to tend to cluster by metastasis site, but metastases from the same patient had very distinct IM expression levels (Fig R4c).

Therefore, these analyses have indeed revealed heterogeneity of cell fractions and IM expression across individual metastases, reinforcing the notion of a variable spatial architecture of the TME. The initial study did not aim the examination of the TME across different metastases, and therefore the number of metastasis from different target organs is limited to draw conclusions. A larger series of metastases (both from different sites and

within a patient) is needed in order to conclude if the TME in metastases from PPGLs simply reflect the metastatic organ site, or mirrors e.g. specific genomic features. Both possibilities have already been described in metastatic breast cancer (De Mattos-Arruda, *et al. Cell Rep.* 2019). When expanding the series, characterization of the TME composition across metastases from individual cases would also be interesting, as it could have implications for immune checkpoint inhibitor therapy because, for example, *CD274* (PDL1) expression was different across metastases from patient A (Fig R4c).

Fig R4. The tumor microenvironment is heterogeneous across metastases. **(a)** Immunescore, **(b)** immune cell levels and **(c)** immunomodulator expression across metastases. Different colors denote metastatic organ site as shown in the legend. 'A' and 'B' indicate metastases from the same patient.

We acknowledge the Reviewer for bringing this important issue up. Although the results obtained from these analyses are promising, they are still based on a very small number of cases, and conclusions have to be taken with great caution. This is the reason why we are not including them in the body of the article.

However, what is important to note is that metastases were only included, with respect to the characterization of the TME, in the analyses represented in Figures 6A, 7A, and ED Fig 10B-C (before ED Fig 9B-C) and 11A (before ED Fig 10A), as it is indicated in the figure

legends. We believe that the inclusion of these data does not bias any of the conclusions presented, but if the Reviewer deems it appropriate, we could remove those data from those particular figures.

7. Just by eyeballing Figure 1, an IHC panel consisting of ATRX, Ki-67 (5% cut-off) and SDHB would potentially be adequate in identifying many of the PPGLs with high TMB/MSI, and probably have high sensitivity for most metastatic cases. Since these markers are routinely available in most pathology laboratories (while CDK1 is not), please comment on the potential of the former. I assume there are no established clinical routine analyses for interrogating TERT expression?

R/ We have evaluated the ability of *ATRX*, Ki-67 and *SDHB* to identify PPGLs with high TMB/MSI by calculating the area under a receiver operating characteristic (ROC) curve (AUC). To perform the analysis, we have used *ATRX* status (*ATRX*-mutated, as no *ATRX* IHC was available for this study), high *MIK67* expression (as we had very few Ki-67 IHC available) - the threshold for high *MIK67* expression was established by the tumor with the lowest *MIK67* expression among those positive for Ki-67 IHC (>5% positive cells) -, and *SDHB* status (*SDHB*-mutated). Only primary tumors with all data available were included for the analysis (n=156). We observed that none of the variables individually was able to identify tumors with high TMB or MSI. When combining the three variables to predict both high TMB and high MSI score, the AUC was 0.64 (95%CI=0.43-0.84) and had low sensitivity (Fig R5a,b).

Fig R5. Receiver operating characteristic curve analysis showing the ability of *SDHB*-mut, *ATRX*-mut and high *MIK67* expression to predict cases with (a) high MSI score and (b) high TMB

Therefore, we believe that the identification of tumors with high TMB or MSI could be more efficaciously performed using an NGS panel rather than the proposed IHC panel. Guidelines have been proposed to assess TMB from NGS panels (Merino *et al. J Immunother Cancer*

2020). In fact, it is stated that less mutated tumors (as would be the case for PPGLs) are the ones that show less variability between panel and WES results. At the moment, there are two targeted NGS panels to assess TMB, which are more cost effective and have a lower turn-around time than WES, and that have been authorized by the FDA (Foundation Medicine FoundationOne CDx test and the Memorial Sloan Kettering Cancer Center MSK-IMPACT). Of course, testing and validation of its utility for PPGLs would be strictly needed. Furthermore, a customized NGS panel sequencing the loci used for MSI score calculation could also be designed.

Thanks to this Reviewer question, however, we have now evaluated the ability to predict mPPGL of high MSI score, high TMB, chr 5 gain and high *CDK1* expression, as well as the ability of the known markers of mPPGL (Krebs cycle gene mutations, *MAML3*-fusion, *TERT*-alt, *ATRX*-mutations and high *MIK67* expression) by calculating the ROC's AUC of each event individually and combining them. Variables were dichotomized and all possible combinations were tested (n=511) in the n=156 tumors in which we had data for all variables. The AUC, 95%CI and sensitivity/specificity per each classifier are now in Supplementary Table 3. The best classifier is the one that takes into account *ATRX*-mutations, high MSI score, high *CDK1* expression and *MAML3*-fusions, showing a 100% sensitivity with the highest ability to predict mPPGL (AUC = 0.902, 95%CI = 0.855-0.948) (Fig R6).

	AUC [95% CI]	P-value	Sensitivity	Specificity
ATRX -mut – high MSI score – high CDK1 expr. – MAML3 -fusion	0.902 [.855, .948]	8.58e-13	100	80.3
ATRX -mut – TERT -alt – Krebs Cycle gene-mut – MAML3 -fusion – high MIK67 expr.	0.793 [.706, .879]	1.83e-7	82.3	76.2
high MSI score	0.819 [.719, .920]	1.29e-8	64.7	99.2
high CDK1 expr.	0.750 [.647, .852]	8.70e-6	64.7	85.2
high TMB	0.688 [.574, .801]	8.26e-4	44.1	93.4
Krebs Cycle gene-mut	0.671 [.558, .785]	2.27e-3	44.1	90.2
TERT -alt	0.642 [.527, .757]	1.15e-2	38.2	90.2
high MIK67 expr.	0.587 [.470, .703]	0.12	20.6	96.7
chr 5 gains	0.568 [.452, .683]	0.23	17.6	95.9
MAML3 -fusion	0.557 [.442, .672]	0.31	14.7	96.7
ATRX -mut	0.555 [.440, .670]	0.33	11.8	99.2

Fig R6. Receiver operating characteristic curve analysis showing the ability of the best classifier for mPPGL (*ATRX*-mut, high MSI score, high *CDK1* expression and *MAML3*-fusion), the ability of the classifier that includes the already known mPPGL markers (*ATRX*-mut, germline Krebs cycle mutations, *MAML3*-fusion and high *MIK67* expression), and the ability of each event independently. The AUC, 95%CI and sensitivity/specificity for the rest of the calculated classifiers is in Supplementary Table 3.

We have now included this analysis into the ms text as follows:

Results section (line 262-272) – “We evaluated the classification power of the potential markers of mPPGL described in this study: high MSI score, high TMB, gain of chr 5 and high *CDK1* expression. We also aimed to assess their power in combination with the already

described markers of mPPGL (germline mutation in Krebs cycle genes, *MAML3*-fusion, *TERT*-alt, *ATRX*-mutation and high *MKI67* expression). The area under the receiver operating characteristic curves (AUC under ROC) of each event individually and all combinations possible (n=511) were computed. The AUC, 95% confidence interval (CI) and sensitivity/specificity per each classifier are in Supplementary Table 3. The best classifier is the one that takes into account *ATRX*-mutations, high MSI score, high *CDK1* expression and *MAML3*-fusions, showing a 100% sensitivity with the highest ability to predict mPPGL (AUC = 0.902, 95%CI = 0.855-0.948) (Extended Data Fig. 7).”

Discussion section (line 439-444) – “Given what has been reported to date, it will be extremely difficult to identify a single molecular marker capable of accurately predicting the metastatic behavior of these tumors. Therefore, it is of particular interest that our study describes a classifier that takes into account only 4 events (*ATRX*-mutations, high MSI score, high *CDK1* expression and *MAML3*-fusions) and achieves a sensitivity of 100%. Further validation will be necessary, but we are, for the first time, looking at a classifier that could identify all patients at risk of metastasis at the time of diagnosis of the primary tumor.”

Methods section (line 719-726) – “The individual variables to be evaluated were dichotomized as previously explained. The total number of possible classifiers was dichotomized by computing all possible combinations with the 9 events to be evaluated (n=511). We set as 1 - when any of the evaluated events included in the classifier was present; 0 - when none of the events was present. This analysis was executed using the dichotomized matrix with the samples for which we had all the data available (n=156). We calculated the area under the receiver operating characteristic curves (AUC under ROC) of each event individually and all combinations possible (n=511).”

Regarding the use of a potential IHC panel proposed by the Reviewer to identify mPPGLs, we could state that according to our data (*ATRX*-mut, high *MKI67* expression and Krebs cycle gene-mut), it would only have a sensitivity of 58.8% and specificity of 86.9%. This information can now be seen in Supplementary Table 3. Therefore, the classifier we propose improves, by far, the ability to predict mPPGL.

Finally, and regarding the last question, the Reviewer’s assumption is correct. We have searched and have been unable to identify any established clinical routine test for interrogating *TERT* expression.

Minor points:

8. What is "normal tissue" in the context of PPGL, do the authors refer to normal adrenal tissue or normal paraganglia? This is crucial information. If normal adrenal medullae were assessed, please comment on the discrepancies compared to paraganglia.

R/ We have included 5 “normal tissues”. Three of them are from the CNIO cohort, and as described in Extended Figure 1B, they are normal adrenal medullae. We have now corrected Supplementary Table 1: instead of “Adjacent Normal Tissue”, now it reads “normal adrenal medulla”. The other 2 tissues labeled as “normal tissue” come from the TCGA project and

we don't know their origin. We agree with the Reviewer that there are clear differences between adrenal medulla and normal paraganglia, but, in fact, these data were not used for any of our analyses. Therefore, our results will remain the same regardless of the origin of the so-called normal tissues. The RNA-Seq data from these "normal" tissues have been deposited in EGA, along with those from our tumor series, since we believe they may be of great use in future studies. If the Reviewer deems it convenient, we can delete this information from the body of the article.

9. Discussion: "...the molecular switches involved in metastatic progression are likely variable and involve non-genetic events." Since the authors are not interrogating the cohort via WGS, but rather WES+RNA seq, there is still a possibility that the events driving metastatic spread dwell outside of the exome.

R/ The Reviewer is right. We have added this information in the discussion. Now, it reads (line 428-432): "Therefore, and even though PPGL tumor formation is clearly driven by germline or somatic mutations, our data suggest that the molecular switches involved in metastatic progression are likely variable and may involve non-genetic events. However, future efforts should also be directed towards performing WGS to capture potential alterations in non-coding regions or large rearrangements".

10. Row 624: "hypermethylation" is misspelled. Amended

11. Figure 6C: "no-metastatic"  "non-metastatic". Amended

Reviewer #2, expertise in bioinformatics and immunogenomics (Remarks to the Author):

Calsina et al. describe the results obtained by the genomic profiling of a large cohort of mPPGLs. They integrate their data with those of the TCGA to compare metastatic vs non-metastatic PPGL. Comparison of MSI score and TBM show some differences between PPGL and mPPGL. Despite the various detailed analyses performed by the group, there is no clear genomic hallmark that predicts metastatic vs non-metastatic diseases. However, the study is of interest to the community since the data generated can be used for secondary studies. Moreover, the reported immune characterization is the main interesting part but could be further expanded.

Major points:

1. The authors mention that they have a good number of tumors ($16+4=20$) for which there is more than one tumor sequenced by WES. However, this is not exploited in the paper. Are these multiregional samples, just replicates or longitudinal samples? Even if the number is small, if they have longitudinal samples the author should try to exploit these data. Even if these cases are multiregional, the evaluation of heterogeneity and subclonal analysis could

help to better elucidate the heterogeneity of this disease. The authors could try to better describe and exploit these tumors.

R/ We agree with the Reviewer that this is a very important point and we are fully confident of the potential information these studies would bring to the PPGL community. In fact, our group is currently developing a new project in which we attempt to collect and evaluate an increased number of patients/samples in order to explore intratumor heterogeneity and evolution with sufficient statistical power. Also, and thanks to Reviewer #1-comment #6, we will now also aim to examine the TME across different metastatic sites.

In this new research line, we will track the evolution of pheochromocytoma by inferring clonal structure in primary-relapse-metastases trios to study the different patterns of metastatic spread (e.g. depending on rapid progression to multiple tissue seeding sites or stable disease). In a preliminary analysis, we have identified that high impact consequence variants (category 7) clones tend to be selected in the metastases (Fig R7), revealing the potential of this study.

Fig R7. Characterization of metastazing clones. Dots symbolize the percentage of the average of each type of clone (determined by the cancer cell fraction value –CCF – calculated as in TRACERx study; non-selected = only present in primary tumor, maintained = present in primary and metastasis with similar CCF, and selected = subclonal in primary ($0.1 < CCF < 0.5$) and clonal in metastasis ($CCF > 0.5$) or not present in primary and subclonal or clonal in metastasis) per each paired tumor. Paired t-test is used to test for differences between tumor clones “not selected” and “selected” in metastasis. **(A)** Clones of variants with low-, moderate- and high- impact consequence, **(B)** Clones of variants with high impact consequence. Clones of high impact consequence are significantly more selected in metastases.

In a similar direction, we also want to decipher the clonal origin of multifocal PPGLs, as has recently been done for small intestine neuroendocrine tumors (Elias, E *et al. Nat Commun.* 2021).

As the Reviewer mentions, the cohort of multiple sequenced tumors per patient is small. Specifically, for 4 patients, we sequenced multiple primary tumors; for 1 patient, we have WES available from the primary tumor and its relapses; for another patient, we sequenced multiple primary tumors and one of its metastases; for 3 patients, we sequenced the primary tumor, its relapse and a metastatic tissue; and for 11 patients we sequenced the primary tumor and it(s) metastasis(es). As the objective of the present manuscript was not directed at exploring the tumor evolution and intratumor heterogeneity, we did not sequence further longitudinal samples nor did we include multi-regional samples from each tumor.

Importantly, we are also planning to delineate the different mutational processes causing chromosomal instability (CIN) in PPGLs by quantifying copy number signatures using the approach developed by our collaborators (Drews RM *et al. Nature* 2022). We aim at exploring the evolution of CIN in PPGLs. However, we need first to adapt the current approach to be able to detect the type of CIN causing the acquisition of each copy number alteration. Unfortunately, the development and validation of this new approach will take longer than the time allotted for this revision to include in the present article.

2. The transcriptomic analysis is based on the integration of different platforms and different dataset using combat. If I understand correctly, the supervised analysis of Fig 4a is performed on the integrated dataset. Could the relatively low number of differentially expressed (DE) genes depend on the over-smoothing induced by Combat? Did the authors try to separately perform supervised analysis between mPPGLs and PPGLs? It seems from Figure ED1b that it would be possible. The validation of ED5a reports just the 27 genes of Figure 4a. What about all the DE genes in this validation dataset? Overall, it seems that the transcriptomic signature is not robust enough and the authors should try to better validate the signature. However, the prognostic role of CDK1 is convincing, but not completely novel.

R/ Integration of genomic data from different studies and platforms increases statistical power, but is often hampered by batch effects. The batch effect adjustment step is essential to eliminate unwanted variation in the data caused by differences in technical factors between batches which can produce false positives in differential expression and therefore erroneous results. It is possible that genes with the lowest variation between conditions might be susceptible not to be found significantly differentially expressed after batch effect adjustment. However, batch effect adjustment methods such as Combat intends to remove batch effects and keep the biological signal in the data; therefore, genes with significant differences can be detected. In this regards, we identified 351 genes DE (FDR<0.05; log₂FC>|0.75|). The selection of the 27 genes was performed taking into account also the DE between metastatic and non-metastatic within each genomic subtype, as stated in methods (line 700-704) as follows: "Differential expression analysis was executed between non-metastatic and primary metastatic tumors within each genomic subtype with DESeq2v1.18.1 using estimate score as a covariate. Best candidates were ranked, and only those with |log₂ fold-change|>0.75 in all analyses and FDR<0.01 in differential expression analysis including all cases were selected." Therefore, the low number of genes shown in the mPPGL-related signature are extremely robust and genotype independent.

Regarding the second question, we have not performed a supervised analysis between mPPGL and PPGLs separately in the two cohorts (CNIO and TCGA) before batch effect correction with Combat. The reasons why we have not proceed to do so are as follows:

- (1) All analyses with RNA-Seq data have been performed using the integrated dataset (Figure 4 through Figure 7); by keeping the integrated cohort from the beginning, we ensured a homogenized data set.
- (2) The use of the integrated dataset allows for greater statistical power

- (3) The TCGA and CNIO cohorts are very different in terms of the number of non-mPPGLs and mPPGLs (the TCGA series is enriched in non-mPPGLs, and the CNIO one in mPPGLs); if the analyses are performed separately, the results could be heterogeneous.

Another concern that the Reviewer is raising in this section is about a DE analysis in the validation. We have now performed this DE in the independent cohort and applied the same filtering criteria as for the discovery cohort. As an exception, we did not take into account the Wnt-signaling subtype to filter for FC (as the validation dataset has only 1 mPPGL and 3 non-mPPGLs from this subtype); in addition, only probes with lower FDR per each gene were selected. As a result from this analysis, we identified 165 DE genes, 7 of which had been identified through the analysis in our cohort, *CDK1* being one of them. We have now highlighted them with a triangle in Extended Data Figure 5a. The representation of these 165 DE genes is in Fig R8a. After a univariate and multivariate logistic regression analysis (Fig R8b), 145 still remained significant as potential markers of risk of metastasis. However, although the multivariate analysis used as a co-variate the genomic subtype, we still could observe a marked associated expression to genotype of some of the genes, which was barely appreciable in our 27 gene signature reported and extracted from our cohort (Fig 4a). As we can observe, some of the genes extracted from this validation cohort show a similar expression pattern between mPPGL and pseudohypoxic non-mPPGLs (more evident with SDHx in the upper right part of the Figure R8a). The reason for this could be that our cohort, apart from being larger than the validation one (n=231 vs 186), has a better representation in the number of tumors belonging to each genomic subtype. Specifically, our cohort has lower number of pseudohypoxic tumors within mPPGLs and a higher number of pseudohypoxic tumors within non-mPPGLs than the validation cohort (60% vs 70%, and 38% vs 33%, respectively). This results in the fact that the 30.3% (44/145) of the genes that appear significant in the DE of the validation cohort have also been previously associated to a certain genotype (reported by Burnichon *et al.* 2011 and/or López-Jiménez *et al.* 2010), while only 1 (*PAH*) of the 27 genes (3.7%) from our signature appear in that list. Accordingly, we have now eliminated *PAH* from the mPPGL-related signature, reducing our signature to a 26-genes signature. Figures 4a, 4b, 4c and ED Figures 5a, 5b, 5c and 5d have been modified accordingly, as well as the text (lines 226 to 236).

Fig R8. mPPGL transcriptomic profile in independent cohort. (a) Gene signature associated with mPPGL. mRNA expression levels of the 165 differential expressed selected genes; tumor behavior, genomic subtype and genotype are depicted in rows; primary tumors appear in columns (n=230). **(b)** Univariate (black) and multivariate (blue) logistic and Cox regression analysis of metastasis risk. Gene expression was dichotomized as follows: for down-regulated genes in mPPGLs, median expression was used as threshold (0 – below the median expression level; 1 – above the median expression level); for up-regulated genes in mPPGLs, high expression levels > than the 3rd quartile (0 – below the 3rd quartile threshold value; 1 – above the 3rd quartile threshold value). Multivariate analysis included as covariate genomic subtype. Only data from primary tumors from non-metastatic and metastatic patients was included.

In this study we have identified 26 genes DE between mPPGLs and non-PPGLs, independent of genotype, 14 of which have value for stratifying patients according to metastatic risk in two different cohorts (ED Fig 5c,d). Given that we have used the largest series of well-characterized cases described to date, we believe that it is difficult to obtain a validation series for this study. We certainly agree with the Reviewer that the signature should be validated in future studies. Toward this goal, we plan to leverage the European Network for the Study of Adrenal Tumors (ENS@T) consortium to conduct a powerful large, multicenter and prospective study of this 14-gene signature. We would also like to take the advantage to validate either the expression of *CDK1* and/or a digital pathology methodology to evaluate CDK1 protein levels as a prognostic factor to be potentially implemented in the clinical setting.

3. The immune landscape is interesting, but it is not related to the metastatic PPGL, rather it confirms the heterogeneity of mPPGLs and the lack of differences. Also, the TME subtypes are loosely associated with the genomics subtypes with the exception of pseudohypoxic in the Fibrotic TME subtype. Figure 7c could be done differentiating metastatic versus non-metastatic.

R/ We believe that there has been some kind of confusion and that the Reviewer was referring to Extended Data Fig 10c (now ED Fig 11e), as figure 7c is related to the expression in different immunomodulatory genes. If so, also the proportion of tumors per TME subtype according to their behavior was already shown in Extended Data Fig 10a (now ED Fig 11a). If we have misunderstood what the Reviewer is referring to, please let us know and we will be happy to follow up.

4. The data of figure 7e showing the greater infiltration of CD8+ T cells in *MAML3*-tumors can be interesting but is limited few cases. Why this was not detected using the Cybersort deconvolution?

R/ CD8+ T cell infiltration in *MAML3* positive tumors in the present manuscript using CIBERSORTx was not shown as we considered that the discovery RNA-Seq cohort had few *MAML3*+ tumors (6% of the whole series) to extract conclusions. In contrast, the FFPE series used to detect by IHC CD8+ T cells was enriched in *MAML3* tumors (7/44, 15,9%) allowing us to have more robust and comparable results.

However, as proposed by the Reviewer, we have now gathered CIBERSORTx results from our cohort and also obtained CIBERSORTx output from an independent cohort (Burnichon *et al.*, 2011). We have compared relative CD8+ T cell levels in *MAML3*-related mPPGLs with other mPPGL of the other genomic subtypes (Fig R9). Although a tendency of a greater infiltration of CD8+ T cells is observed in *MAML3* tumors, no statistically significance was reached.

Fig R9. Standardized CD8+ T cell levels estimated with CIBERSORTx in mPPGLs from our cohort and independent cohort according to genomic subtype.

It is also important to mention that there are important caveats when using the presented bulk RNA-Seq data for immune phenotyping. For most tumors included in the CNIO series, tumor selection (>60% cancer cells) was performed by a pathologist, as well as for all TCGA samples, thus potentially removing the most immune-infiltrated tumor regions from analysis. Figure R10 are representative images of a CD8 IHC in a *MAML3* tumor. As the reader can appreciate, region IV has a much elevated number of CD8+ T cells in comparison with region II (\approx 5-fold). Whereas with the quantification of CD8 IHC we took into account the whole slide, some of the tissues included in the bulk RNA-Seq were selected to have as much tumor content as possible. Therefore, if hypothetically region II was selected for RNA-Seq analysis, the number of these cells would have been underestimated. In fact, CIBERSORTx output was only partially correlated with IHC scoring ($r=0.50$; $P=0.025$).

This does not mean that the conclusions that we present in this ms are not reliable, but what is true is that, most probably, the differences between the immune cell types detected by CIBERSORTx without having made a selection of tumor cells would have been more significant. Meaning that the levels of tumor-infiltrating leukocytes in certain tumors (most certainly *MAML3*-related ones) would be higher. This issue was also discussed by Thorsson *et al.* (*Cancer Cell.* 2019), who presented the immune landscape in all TCGA samples.

Fig R10. Representative images of an IHC for CD8 of a *MAML3*-tumor. (I) Whole slide section; (II) and (IV) close-up of two distant regions of the slide indicated in (I) with a red rectangle – scale bar=200 μ m; (III) and (V) close-up from previous images in region indicated with black rectangle – scale bar=100 μ m.

5. Related to the previous point, the authors describe the TME classification of PPGLs using the large dataset and the integrated transcriptomic. However, there is no downstream analysis of these clusters. Are there any genomics events associated with these TME clusters? Is there any enrichment of genomics drivers in one cluster vs the other? What about the MSI score and TBM that you have reported in terms of the genomics clusters?

R/ We have evaluated the association between TME subtypes and the 9 genomic events that we have assessed as markers of mPPGL (Reviewer#1, question#7) (Fig R11).

Fig R11. Association analysis between the TME subtypes (as in the manuscript) and the nine genomic alterations shown in the top-axis of the figure. The length of the bars represents the % of tumors with each of the top-axis alterations in each TME subtype; the scale of the % for each alteration is the same – as shown on the y-axis scale for “high *CDK1*”. Black bars are present for alterations with significant association after applying a Chi-squared test; grey bars are present for alterations with non-significant association with TME subtypes.

We had already observed an enrichment of the pseudohypoxic cluster in subtype F (before Extended Data Fig. 10c, now ED Fig 11e), which here is observed also for those tumors with germline Krebs cycle mutations ($P=1 \cdot 10^{-4}$). Bagaev *et al.* (*Cancer Cell*, 2021) reported high chromosomal instability (CIN) in melanoma subtypes D and F, and we believe that the high MSI score detected in subtype F ($P=0.047$) could be related to this CIN. Bagaev *et al.* also showed the highest proliferation rate in subtype D; accordingly, we have observed the highest proportion of tumors with high *MIK67* in subtype D ($P=0.001$). Finally, high *CDK1* expression was associated with subtype D and F ($P=5 \cdot 10^{-4}$). *MAML3*-fusions, *ATRX*-alt, high TMB, *TERT*-alt and chr5 gains did not show significant association with any TME subtype.

Regarding the existence of an enrichment of genomic drivers in one group vs the other, we believe that this was already explored in the initial version of our ms. This is represented in Extended Data Fig. 11e (before ED Fig 10c), and in the ms text (lines 369-379) it reads: “TME subtype proportions also differed between immune subtypes (Extended Data Fig. 11d) and PPGL genomic subtypes (Extended Data Fig. 11e). The pseudohypoxic cluster was mainly enriched in the fibrotic subtype, characterized by high vascularity and an immunosuppressive phenotype and as recently reported by Celada *et al.*; in contrast, the kinase signaling cluster was highly represented in the immune-enriched non-fibrotic subtype, reflecting the influence of the genomic subtypes on TME. These differences were also evident when comparing Fges scores between genomic subtypes (Extended Data Fig. 11f). We observed higher representation of a Th1 signature and checkpoint molecules in the kinase signaling tumors, and the pseudohypoxic tumors were characterized by higher signature Fges scores related to angiogenesis and fibrosis (cancer-associated fibroblasts, matrix remodeling, endothelium), as already reported.”

Concerning the association of differential TMB across genomic subtypes, we believe this was also explored in the original version of our article. In Extended Data Fig. 3a,b we plot TMB according to genomic subtype, and the text reads (lines 158-159): “We also observed a significantly different TMB among genomic subtypes, with the Wnt-altered subtype having the highest values, regardless of tumor behavior (Extended Data Fig. 3a,b)”. We have now

explored the MSI score in the different genomic subtypes and cannot see any difference within the groups (Fig R12).

Fig R12. MSI score among PPGL tumors with different genomic subtype in non-metastatic tumors, and in primary tumors from mPPGLs.

In our opinion, the new analyses do not add substantial information to our article and, therefore, we have not included it in the body of the ms. However, if the Reviewer considers it interesting, we would proceed to add it, as well as to perform association analyses with any other genomic events that may be considered of interest.

6. If the authors want to describe an immune landscape of mPPGLs, that part of the paper completely lacks the study of the neoantigens. I suggest, in addition to the TMB, characterizing the neoantigen load in the four TME subtypes, and interpreting immunoeediting events.

R/ We thank the Reviewer for this important suggestion. We have now characterized the neoantigen load per sample and associated it with the four TME subtypes. We did not observe significant differences in neoantigen load or TMB between TME subtypes (Fig R13a,b), finding expected due to their highly significant correlation (Fig R13c). However, a trend towards a higher load of both was noted in IE/NF and D. A higher mutational load in IE/NF and a higher chromosomal instability also in D had already been described by Bagaev *et al.* (*Cancer Cell.* 2021). Therefore, we believe that these trends will be confirmed if a larger series with greater representation of these two subtypes is constructed.

Fig R13. (a) Neoantigens load and (b) TMB across the different TME subtypes in primary PPGL tumors. P shown in the figure corresponds to a Krustal-Wallis test. (c) Correlation between TMB and neoantigen load. Pearson's correlation coefficient is shown

We have now included this information in the ms (line 368-369) as follows: “We also observed a trend towards TME subtypes being associated to neoantigen load and TMB (Extended Data Fig. 11b,c)”. The methodology for neoantigens identification was also included in the methods section (line 775-787): “Manually curated somatic SNVs and indels from the CNIO and TCGA cohort were annotated with VEP v106 using the required and useful VEP options specified in the pVACtools documentation. Coverage information for variant sites was retrieved from tumor and normal bam files using bam-readcount (Larson. genome/bam-readcount. GitHub (<https://github.com/genome/bam-readcount>)). For this purpose, the Docker 1.1.1 image of mgibio/bam_readcount_helper-cwl was used and the collected data were added to the annotated VFC files with the vcf-readcount-annotator from the VAtools package (<http://vatools.org>). HLA class I typing was performed on each sample with POLYSOLVERv4. pVACseq from the pVACtools package version 3.0.2 was then used for epitope prediction. The epitope length was set to 8-11 and the prediction algorithms were all those available for MHC Class I (NetMHCpan, NetMHC, NetMHCcons, PickPocket, SMM, SMMPMBEC, MHCflurry and MHCnuggetsI). The epitopes provided by pVACseq in the filtered files, which are those passing binding affinity and sequence-based thresholds were the ones used to establish the neoantigen load per sample.”

Thanks to this Reviewer's point, we have also evaluated the neoantigens load in *MAML3*-tumors compared to other PPGLs. As reported by Yang *et al.* (*Nat Med.* 2019), gene fusions are a source of immunogenic neoantigens. With this new analysis, now, we can corroborate this finding in our *MAML3*-fusion tumors. These new data has been incorporated in a new figure (Figure 7e), and the results now reads (line 392-393): “A higher neoantigen load was also observed in *MAML3*-tumors compared to all other tumors (MWW test: P=0.0067; Fig. 7e).”, as well as it has been added in the discussion (line 468).

7. Fastq files have been deposited on EGA (but still not available). If this study wants to complements the TCGA for mPPGLs, the authors should definitely share level 3 data (count matrices, MAF files, copy number segment files).

R/ The Reviewer is right and level 3 files are much more useful resource than raw files. Therefore, we have now added to the same EGA submission the expression data (count matrix), and the VCF files with the variant calling results (for both SNV/INDELs and CN).

Just to clarify, Fastq files were already available at the time of ms submission (now, also level 3 files) for Reviewers to consult through a specific reviewer account that EGA had to provide to the journal editor upon request. Please, let us know if we can facilitate this process in any way.

Reviewers' Comments:

Reviewer #1:

Remarks to the Author:

I would like to thank the authors for the clear and elaborate revision. I agree with the responses to my previous queries and the changes made to improve the content, and have no further issues to report.

Reviewer #2:

Remarks to the Author:

Calsina et al. have performed further analyses and clarified some of the points raised in the previous submission.

I believe the paper represents an excellent contribution to understanding the molecular features of mPPGL, and I am satisfied with this revision.